# Classification and Evaluation of Concepts for Improving the Performance of Applied Energy System Optimization Models

**Karl-Kiên Cao \*** , **Kai von Krbek, Manuel Wetzel, Felix Cebulla †** and **Sebastian Schreck †**

Department of Energy Systems Analysis, Institute of Engineering Thermodynamics,
German Aerospace Center (DLR), Pfaffenwaldring 38–40, 70569 Stuttgart, Germany; kai.krbek@dlr.de (K.v.K.);
manuel.wetzel@dlr.de (M.W.); felix.cebulla@googlemail.com (F.C.); schreck.sh@gmail.com (S.S.)
\* Correspondence: karl-kien.cao@dlr.de; Tel.: +49-711-6862-459
† Former members.

**Abstract:** Energy system optimization models used for capacity expansion and dispatch planning are established tools for decision-making support in both energy industry and energy politics. The ever-increasing complexity of the systems under consideration leads to an increase in mathematical problem size of the models. This implies limitations of today's common solution approaches especially with regard to required computing times. To tackle this challenge many model-based speed-up approaches exist which, however, are typically only demonstrated on small generic test cases. In addition, in applied energy systems analysis the effects of such approaches are often not well understood. The novelty of this study is the systematic evaluation of several model reduction and heuristic decomposition techniques for a large applied energy system model using real data and particularly focusing on reachable speed-up. The applied model is typically used for examining German energy scenarios and allows expansion of storage and electricity transmission capacities. We find that initial computing times of more than two days can be reduced up to a factor of ten while having acceptable loss of accuracy. Moreover, we explain what we mean by "effectiveness of model reduction" which limits the possible speed-up with shared memory computers used in this study.

**Keywords:** energy systems analysis; energy system optimization models; linear programming; mathematical decomposition; model reduction; REMix

## 1. Introduction

### 1.1. Motivation

Deregulation and growing decentralization have led to an increasing complexity of energy systems. Given the envisaged creation of a common European energy market and the transformation of energy supply towards sectoral coupling and electricity generation from variable renewable energy sources (vRES), this trend can be expected to continue.

In this context, new energy policies are often investigated with the help of linear optimization models [1]. However, the increasing complexity of the system to be modelled results in energy system models that quickly reach their limits in terms of memory demand and reasonable computing time. Existing and especially future research questions in the field of energy system analysis can thus only be addressed to a limited extent. In applied studies, this challenge is tackled with different strategies. Out-of-the-box solutions that enable the use of massively parallelized high performance computers are not available, since therefore additional knowledge, e.g., about the matrix structure of the mathematical optimization problem is necessary. Therefore, the majority of currently applied speed-up strategies still

rely on the application of commercial optimization software executed on shared memory hardware. However, the implementation costs and not the effectiveness often dominate the decision for an appropriate performance enhancement approach. In addition, the heterogeneity of applied strategies results in the fact that the comparability of model-based scenario studies is more difficult and the trade-off between implementation costs and achievable performance is often unknown. Since the used models show similarities in essential characteristics (e.g., with regard to fundamental equations or applied solver software packages), it can be assumed that effective speed-up strategies for energy system models are transferable.

Therefore, this article presents a systematic evaluation of such strategies. The characterization of the discussed linear optimization models, which are referred to as "Energy System Optimization Models" (ESOMs), is followed by a categorization and a qualitative description of known approaches for shortening computing times. Subsequently, the implementation for a selection of performance enhancement approaches is introduced and the framework for the conducted benchmark analysis is presented. Finally, an outlook on further possibilities on the reduction of computing time in ESOMs is given.

*1.2. Energy System Optimization Models: Characteristics and Dimensions*

In the context of energy systems analysis a broad spectrum of research questions is addressed by ESOMs to support decision making in both energy politics and energy industry. In particular, this concerns the development of future strategies such as energy scenarios for mitigation of climate change [2] or fundamental analyses of electricity markets [3] and investment planning by system operators [4,5]. Therefore, the objective of the associated optimization problems (OPs) is either the optimal operation or the optimal configuration of the analyzed system which consist of a diverse set of technologies. With regard to electricity generation, the former is originally known as Unit Commitment (UC) or Economic Dispatch (ED) problem [6], while the latter is referred to as Generation Expansion Planning (GEP) [7]. If these problems are resolved on the spatial scale, the consideration of transport infrastructures, such as high voltage transmission grids, and thus modeling of multi-area OPs becomes relevant. Typical examples are Optimal Power Flow (OPF) problems [8] on the operational side and Transmission Expansion Planning (TEP) [9] on the configurational side.

Furthermore, due to the increasing relevance of renewable energy sources in todays and future energy systems, also the evaluation of strategies which make use of electricity storage facilities to integrate fluctuating power generation becomes more and more important [10].

The problems addressed by energy systems analysis are typically combinations of the above mentioned aspects which result in integrated bottom-up models that differentiate three major scales: technologies, time and space. Table 1 shows these scales together with their characteristics for exemplary applications. Two kinds of characteristics are distinguished here. While the descriptive characteristic is related to the description of the underlying real world problem, the model characteristic refers to the way how this problem is translated into a mathematical model formulation.

**Table 1.** Characteristics of Optimizing Energy System Models.

| Dimension | Model Characteristic | Descriptive Characteristic | Example | |
|---|---|---|---|---|
| **Time** | Set of time steps | | Short-term (sub-annual operation) | Long-term (configuration/ investment) |
| | | Temporal resolution Planning horizon | hourly one year | each 5 years from 2020 until 2050 |
| **Space** | Set of regions | Spatial resolution Geographical scope | Administrative regions (e.g., NUTS3 [11]) European Union | |
| **Technology** | Variables and constraints per technology | Technological detail | Consideration of start-up behavior, minimum downtimes | |
| | Set of technologies | Technological diversity | Power and heat generation, transmission grids and storage facilities | |

Depending on the application, the three dimensions are differently pronounced or resolved in energy system analysis. For example, on the one hand, ESOMs are strongly spatially resolved with the aim of cost-optimized network expansion planning by TEP. On the other hand, also the temporal resolution becomes important as soon as a study tries to capture the variability of power generation from renewable energy sources. However, formulating a mathematical model with these characteristics usually results in coupling of time, space and technology among each other. Even more importantly, the need of addressing flexibility demands in future energy systems [12] also leads to couplings within these dimensions. In particular, these couplings are caused by temporally shifting of generation and consumption with storage facilities or demand side management measures which links discrete points in time, by power exchange over transmission grids that results in an interdependency of regions as well as by cross-sectoral technologies such as combined heat and power (CHP) plants.

*1.3. Challenges: Linking Variables and Constraints*

One substantial common characteristic of optimization models, we refer to as ESOMs, is the use of a cost-based objective function conjunction with a power balance equation. For example, Equations (1) and (2) are typical for ED problems (to better distinguish model parameters and variables, in the following, variables are denoted in bold):

Objective function:

$$Minimize : \sum_{t \in T} \sum_{n \in N} \sum_{u \in U} c(t, n, u) \cdot \boldsymbol{p}(t, n, u) \tag{1}$$

Subject to:

$$\sum_{u \in U} \boldsymbol{p}(t, n, u) = \boldsymbol{d}(t, n)$$
$$\forall t \in T, \ \forall n \in N, \ \boldsymbol{p}(t, n, u) \geq 0 \tag{2}$$

where: $\boldsymbol{p}$: (activity-) **variable** of total power supply, c: specific costs, $\boldsymbol{d}$: power demand, $T$: set of time steps, $N$: set of modeled regions and $U$: set of technologies.

Although different ESOMs consist of a large variety of further constraints, such as capacity-activity, flow or security constraints, they share another similarity concerning the structure of the coefficient matrix **A** of the appropriate linear program (Figure 1).

The abovementioned interdependencies of time, space and technologies translate either into linking variables or linking constraints. Both are characterized by the fact that they prevent the OP from being solved by solving independent sub-problems (indicated by the colored blocks in Figure 1). In this context, we refer to the corresponding OPs to be monolithic.

From an applied point of view, linking means, for example, that for a selected time frame the dispatch of reservoir power plants cannot be determined without the information about the storage

level. However, the storage level of the actual time frame also relies on the dispatch of previous points in time.

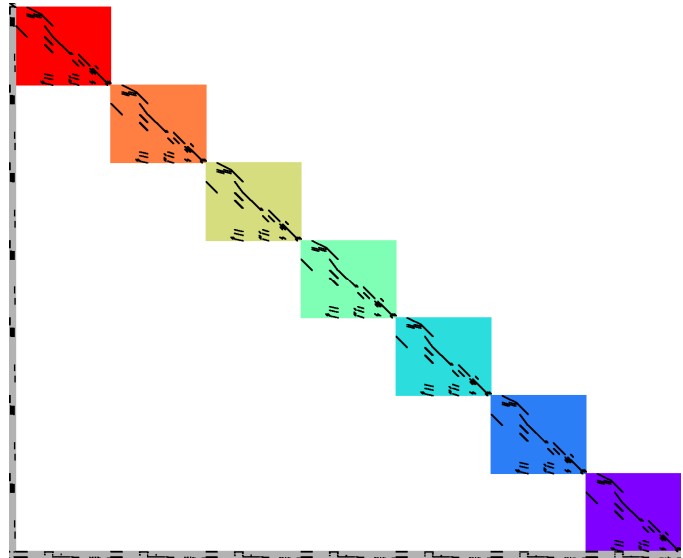

**Figure 1.** Non-zero entries (black dots) in an exemplary coefficient matrix A of an integrated Energy System Optimization Model (ESOM) with linking variables (grey area at the left), linking constraints (grey area at the bottom) and independent blocks (colored blocks).

In this context, variables that occur simultaneously in several equations are generally referred to as linking variables (or sometimes complicating variables). Provided that an appropriate permutation is given, as shown in Figure 1, linking variables appear as vertical lines of non-zero entries in the coefficient matrix. With regard to the temporal scale, representatives of linking variables in ESOMs appear in expansion planning problems as the appropriate investment decision variables (e.g., opposed to activity variables) are not defined for each time step of the operational time horizon. This is illustrated by inequality (3) which is defined for each time step $t$, but the variable $\boldsymbol{I}$ stays the same for each $t$.

Capacity-activity constraint:

$$\boldsymbol{p}(t, n, u) \leq P(n, u) + \boldsymbol{I}(n, u)$$
$$\forall t \in T; \forall n \in N; \forall u \in U$$

(3)

where: $\boldsymbol{I}$: variable of capacity expansion and $P$: existing capacity

In contrast to linking variables, horizontal lines of non-zero entries in the coefficient matrix indicate linking constraints (Figure 1), sometime referred to as complicating constraints. For example, fuel availability constraints, such as used for modeling biomass fired power plants, typically define a temporally non-resolved value as an annual limit. To ensure that the total fuel consumption within the operation period stays within this limit, a linking constraint couples the involved variables:

Fuel-availability constraint:

$$\sum_{t \in T} \sum_{u \in U_{Bio}} \boldsymbol{p}(t, n, u) \cdot \frac{1}{\mu(u)} \leq F(u)$$
$$n \forall N; \ U_{Bio} \subset U$$

(4)

where: $F$: available fuel, $\mu$: conversion efficiency and $U_{Bio}$: set of biomass power plants.

## 2. State of Research

### 2.1. Classification of Performance Enhancement Approaches

We distinguish two methodological layers for approaches to enhance the performance of an ESOM (Figure 2). On the one hand, in the technical layer measures are emphasized that can be taken on the solver side in order to generally solve an OP. Thus, it concerns all methods that are applied in a solver package, such as CPLEX, Gurobi, Xpress or MOSEK, whether it is a specific implementation of solution algorithms or the tuning of the same by an appropriate parameterization. On the other hand, the conceptual layer refers to the translation of a real world problem into an OP. This means, for example, that several possibilities exist on how to address a research question with different model formulations. Model-based measures to improve the performance of an ESOM, thus rely on specific domain knowledge provided by developers of ESOMs. This refers to both the treatment of data in order to reduce the amount of data used in the model as well as the application of heuristics and model-based decomposition methods. In the following, we discuss the state-of-research with regard to model reduction, heuristics and mathematically exact decomposition methods applied to the time, space and technology dimension in ESOMs. Although solution algorithms such as Interior point are applied, we do not focus on improvements on the algorithm side (technical layer). This means that also meta-heuristics like particle swarm optimization or genetic algorithms are not considered.

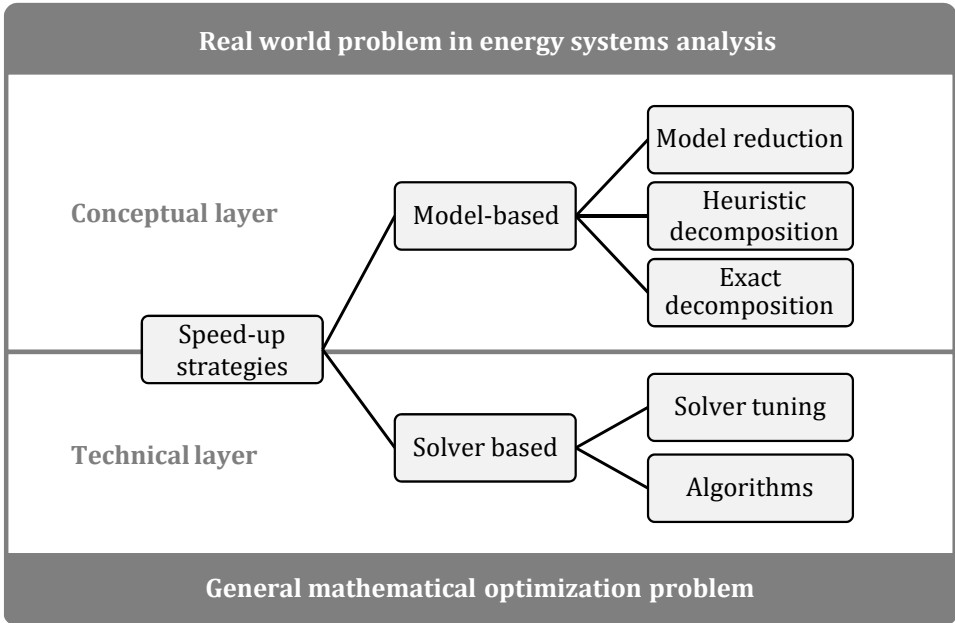

**Figure 2.** Classification of performance enhancement approaches.

### 2.2. Model Reduction

Model reduction approaches are very common since they are effective due to the reduction of the total size of the OP (less variables and constraints). Furthermore, they are also implicitly applied to ESOMs, for instance, due to limited input data access. Thus, these approaches usually manipulate input data in a pre-processing step, instead of changing the way how an ESOM is solved. Based on the treatment of available data we distinguish two forms of model reduction techniques: (i) slicing and (ii) aggregation.

#### 2.2.1. Slicing

Slicing approaches translate into focusing to a specific sub-problem by ignoring existing interdependencies or considering only a part of the input data that could be used. This means,

for example, excluding technologies such as CHP plants from a model [13] or ignoring power exchange beyond neighboring regions on the spatial side [14]. Regarding the temporal dimension, analyses are conducted only for a specific target year [15] or time-slices are selected [16]. These sub-sets represent either critical situations, such as the peak load hour, or typical time periods which are supposed to be characteristic for the entire set of operational time steps. By this means, slicing approaches can lead to significant deviations of results compared to the global optimum of the full OP as they do not ensure that all relevant information within the available data is captured. However, if for the selection of specific slices a pre-analysis is conducted, we do not refer to this process as simple slicing since it aims to take into account all input data. This is rather typical for aggregation approaches. Therefore, they reduce the input data set in a way that relevant information is maintained as far as possible. In the context of ESOMs, aggregation can also be described as coarsening of resolutions for each of the characteristic model dimensions.

### 2.2.2. Spatial Aggregation

The treatment of large, spatially explicit data sets is a common challenge in the context of power network analysis. However, corresponding to the area of responsibility of system operators, methods for power networks were developed to study certain slices of the entire interconnected network. The objective of these traditional network reduction techniques is therefore to simplify the neighborhood of the area of interest by the derivation of network equivalents based on given power flows. These equivalents, such as derived by Ward or Radial Equivalent Independent (REI) methods, represent the external area which is required to show the same electrical behavior as the original network [17]. In the case of Ward equivalents, the networks' nodal admittance matrix is reduced by Kron's reduction [18]. In contrast, however, the REI procedure applies a Gaussian elimination to external buses. Power injections are preserved by aggregating them to artificial generators which are connected to a representative, radial network which is referred to as REI.

The principle of creating network equivalents is also applicable to ESOMs, although their scope is rather the interaction of different technologies than the exclusive assessment of stability or reliability of electrical networks. Recently, Shayesteh et al. [19] adapted the REI approach to use-cases with high vRES penetration. However, this step of creating aggregated regions for a multi-area ESOM needs to be preceded by a partitioning procedure which allows for defining of regional clusters. In general, the clustering algorithms, such as k-means, group regions or buses with similar attributes together. In [19] the admittance between two buses is used to account for strongly connected regions. Opposed to this, Shi and Tylavsky [20] as well as Oh [21] derive network equivalents based on reduced power transfer distribution factor (PTDF) matrices which rely on the linearization of certain system operating points.

Despite the availability of a broad spectrum of sophisticated aggregation techniques, in the context of energy system analysis, the applied literature is governed by simple spatial aggregation approaches. In particular, they are usually characterized by a summation of demand and generation capacities, whereas intra-regional flows are neglected and regions are grouped based on administrative areas, such as market or country borders [15,22,23]. Reasons therefore are, on the one hand side, the availability of required, large data sets of spatially explicit data for the broad diversity of technologies, such as potentials and existing infrastructure. On the other hand, the majority of network equivalents are based on pre-computed system states of the spatially highly resolved model, for example, a solved power flow study. This in turn requires the application of nested approaches (Section 2.3), where first simplifications to other scales of an ESOM are required in order to obtain the power flows of the entire network. By this means, reasonable simplifications are the use of time-slices in form of operational snapshots and the summation of power supply from all generation technologies.

Nevertheless, concerning scenarios of the European energy system Anderski et al. [24], as well as Hörsch and Brown [25] take a step towards improved methodologies regarding aggregation of spatially highly resolved data sets. Both use power demand as well as installed generation capacities

as attributes for state-of-the art clustering algorithms. However, while in [24] PTDF-based equivalents are built, the authors in [25] apply a more or less straight forward process for creating spatially aggregated regions.

### 2.2.3. Temporal Aggregation

Temporal aggregation refers to representative time periods or the process of down-sampling data derived from a highly resolved initial data set. Down-sampling is a method where time series based input data is coarsen to a lower temporal resolution (e.g., by averaging from 1-hourly to 6-hourly). In ESOMs, down-sampling typically affects demand profiles (e.g., electric or heat load) and the power feed-in from vRES. Although the approach is an effective way to reduce computing times—Pfenninger [26], for example, shows a reduction of computing time by up to 80% (scenario 90% 2014)—the method is rarely applied. This is due to the claim to capture the dynamics of variable power provision from renewable energy technologies. By this means, ESOMs typically rely on their highest resolved data and often use hourly input [27]. Exceptions can be found in studies that analyze the impact of different temporal resolutions in unit commitment approaches, e.g., in Deane et al. [28] (5, 15, 30 and 60 min) or in O'Dwyer and Flynn [29] as well as in Pandzzic et al. [30] who both compare a 15 min resolution with hourly modeling.

More common is the combination of down sampling and the selection of representative time periods, such as applied in [31] or [32]. Representative time periods are intended to illustrate typical or extreme periods of time. These time intervals are then weighted to derive the overall time horizon, e.g., one year. Moreover, also challenges exist to account for the chronological relationship between hours which in particular becomes important if time-linking constraints are incorporated in an ESOM. One approach to tackle this issue is presented by Wogrin et al. [33] who define transitions between system states derived by applying a k-means-like clustering algorithm to wind and demand profiles. As stated in [26], the selection of time-slices is either based on a clustering algorithm, such as *k*-means [34], hierarchical clustering [35], or simple heuristics [36].

While temporal aggregation is an effective method to reduce computing times, it is not always clear which error is introduced by it. This issue has been tackled by a number of recent papers, such as Pfenninger [26], Haydt et al. [37], Ludig et al. [31], and Kotzur et al [38]. The studies unanimously highlight the rising importance of high temporal resolution with increasing vRES share. The authors also state that there exists no best practice temporal aggregation and emphasize that it strongly depends on the modeling setup. For instance, Merick [39] recommends ten representative hours for robust scenarios when only variable demand is considered. This number, however, increases significantly when vRES and especially several profiles per technology are taken into account. With regard to representative days, he finds that the number of 300 is appropriate. This represents a clear difference compared to the sufficient number of six representative days resulting in [35]. Nahmmacher et al. [35] use the same clustering technique, but assess model outputs, such as total system costs, rather than the variance of clustered hours of the input time series.

### 2.2.4. Technological Aggregation

We define technology resolution as the abstraction level in a modeling approach to characterize the technologies relevant for the analysis. In this context, it can be stated that the higher the abstraction level, the better the performance of an ESOM. This applies to both the aggregation of input data and the mathematical model of a particular technology. The former, for example, refers to the representation of generation units (electricity, heat, fuels) or flexibility options (e.g., grid, storage). More precisely, classifications of power plant types can be based on several attributes such as rated power, conversion efficiency, and fuel or resources type. Technological resolutions therefore range from very detailed modeling of individual generation units [40] to general distinctions based on fuel consumption and resource [41]. However, the methods for deriving appropriate classifications or aggregations are rather based on simple grouping of attributes than on specific clustering algorithms.

Moreover, the classification of technologies is strongly connected to the mathematical description since physically more accurate models typically require more detailed data. In this regard, a broad body of literature investigates the necessary technological detail for power plant modeling. Often, these analyses compare simplified linear programming approaches (ED) with more detailed mixed integer linear programming (UC) models for least cost power plant dispatch. As a result, such studies assess differences in power plant dispatch (e.g., in [42–45]) and, additionally, highlight effects on resulting metrics (e.g., storage requirements in [46] or marginal prices of electricity generation in [47,48]).

The same applies to transmission technologies where Munoz et al. [49], for instance, study modeling approaches (discrete vs. continuous grid capacity expansion) and their effects on the total system costs. Also technological classifications can be made for different voltage levels or objectives of grid operation (e.g., transmission or distribution). Regarding mathematical models, resolutions range from detailed, nonlinear AC-power flow over decoupled and linear DC-power flow to simple transshipment or transport models [50].

## 2.3. Heuristic Decomposition and Nested Approaches

Although mathematical exact decomposition techniques (see Section 2.4) could be interpreted as nested approaches, in this section, we explicitly refer to methods that usually find near-optimal solutions rather than a theoretically guaranteed exact optimum. In this context, nested approaches are used as a synonym for heuristics. In contrast to meta-heuristics, this concerns methods that imply modifications of the ESOM regarding the conceptual layer and thus base on the same mathematical solver algorithm. In general, nested approaches are built on model reduction techniques (see Section 2.2). Therefore, combinations of several reduced instances of the same initial ESOM (original problem) are usually solved sequentially. This means, that after the solution of the first reduced model is obtained, certain outputs are used as boundary conditions (e.g., in the form of additional constraints) for the following model(s) to be solved.

As mentioned above, ESOMs have linking constraints or variables that globally link points of one dimension. These characteristics are crucial for the decomposition of an OP into smaller instances of the same problem, regardless of whether it should be solved by an exact decomposition (see Section 2.4) or heuristic approach. Often this is intuitively done by the application of nested performance enhancement methods where linking variables, such as power flows or endogenously added capacities are used to interface between the different reduced models.

In the literature, a wide range of examples for the applications of nested performance enhancement approaches exists. For instance, Romero and Monticelli [51] propose an approach for TEP where they gradually increase the technological detail starting with a simple transport model, and finally taking into account Kirchhoff Voltage Law constraints as in a DC-power flow model.

With regard to the spatial scale, one methodology can be described as "spatial zooming", which is similar to the classical methodology applied for power network analysis (see Section 2.2.2). Possible implementations can look like as follows: First a large geographical coverage is considered in a coarse spatial resolution to study macroscopic interdependencies. In a second step, these interdependencies, such as transnational power flows, can be fixed in order to conduct a detailed analysis of the region of interest [52]. In [53] the spatial dimension is simplified by the derivation of network clusters, while for the solution of the original problem a selection of binary variables related to pipelines and suppliers is restricted.

Comparing the different reduced models used in a nested approach, typically, a decrease of resolution on one scale is accompanied by an increase on another. In this regard, one common approach is decoupling investment decisions by "temporal zooming". First, a power plant portfolio is developed over the analyzed planning horizon using a simplified dispatch model and pre-defined time-slices to simulate the planned operation. In order to check whether the derived power plant portfolio performs well for a selected target year, UC constraints are added and capacities are fixed in the subsequent model run(s) [13,43,54]. Babrowski et al. call a similar method "myopic approach" [55]. In this case,

for each year of the planning horizon a model run is performed, whereas the resulting generation expansion is taken as an offset of installed power generation for the subsequently analyzed target year.

In applied energy system analysis, ESOMs often need to consider large sets that represent the temporal scale (i.e., time series of 8760 h) in order to capture the variability of vRES [26], rather than high resolutions on the technological or spatial scale. In the following, we therefore introduce two heuristic methods for this particular dimension in detail.

### 2.3.1. Rolling Horizon

Although the definition of nested approaches does not perfectly fit to rolling time horizon methods, we introduce these heuristics as a preliminary stage to temporal zooming (see Section 2.3.2). The general idea behind rolling horizon methods is to split up the temporal scale (temporal slicing) into smaller intervals to obtain multiple reduced ESOMs to be solved sequentially. In particular, these methods are used for two reasons. One is to account for uncertainties by frequently updating limited knowledge concerning the future. This applies, for instance, to forecasts of load or electricity production from renewable energy sources. Although the main principles of a rolling horizon approach apply to both operational and investment planning, in the following we mainly refer to the former, the rolling horizon dispatch. Therefore, a typical application is short-term scheduling of power systems with a high penetration of renewables [56–58].

The other purpose of implementing a rolling horizon approach to an ESOM is the premise that the total computing time for solving individual partial problems stays below the one for obtaining a solution for the original problem. Marquant et.al [59] report of a wide variety of achieved speed up factors ranging from 15 up to 100.

An optimal number of time windows usually exists depending on the model size, since the computational overhead for creating reduced models increases with the number of intervals. Furthermore, the planning horizon of an individual time window usually includes more time steps than necessary for the partial solution. In the context of energy system analysis, this overlap is important to emulate the continuing global planning horizon. Especially the dispatch of seasonal storage units is strongly affected by this as, without any countermeasures, it is more cost-efficient to fully discharge the storage until the end of an operational period. Also time-linking variables and constraints, such as annual limits on emissions, can barely be considered in this way since global information regarding the temporal scale can only be roughly estimated for each time window. For this reason, inter alia indicated by a trend to overestimate the total system costs [59], the aggregation of interval solutions does not necessarily end up at the global optimum of the original problem.

### 2.3.2. Temporal Zooming

Concerning their capability to improve the performance of an ESOM, rolling horizon approaches have one particular disadvantage. Since each partial solution is updated by a subsequent one, the reduced ESOM instances are sequentially coupled. This prevents parallel solving.

The heuristic, we refer to as temporal zooming, overcomes this issue and allows for solutions closer to the exact optimum of the original problem. Therefore, the rolling horizon approach is adapted in the following way. In a first step, time-linking information is gathered from the solution of an additional ESOM instance which is reduced on the temporal scale. But, in contrast to the reduced ESOMs which consider specific intervals within the full operational horizon, the temporal resolution is down sampled. This in turn allows optimizing the dispatch of the original problem for the full planning period. Values of variables from this first model run can subsequently be used to tune the consideration of global time-linking variables and constraints within the intervals. Despite the need for an additional model run, total computing times for obtaining a final solution can be expected to be at least competitive compared to rolling horizon approaches. This is due to the fact that overlaps are not required and the temporally sliced ESOMs can be solved in parallel.

## 2.4. Mathematically Exact Decomposition Techniques

Decomposition approaches are a well-known instrument for reducing the computing time in OPs. In this case, an OP is broken down into interlinked partial problems. With regard to the structure of the OP's coefficient matrix, the decomposition can be exploited for the creation of individual blocks. Ideally, block structures with globally linking variables or constraints can be isolated from the sub-problems, making them solvable independently of each other, for example in parallel.

Despite this similarity to nested approaches, such as temporal zooming, the crucial difference concerning exact decomposition techniques is the theoretically proven guarantee to find the optimal solution of the original problem [60]. However, this typically requires an iterative solution of partial problems. Therefore, it can be stated, that compared to nested approaches, decomposition techniques provide the best accuracy possible, but at the expense of additional computing time.

### 2.4.1. Dantzig-Wolfe Decomposition

In particular, approaches that can treat linking constraints are Dantzig-Wolfe decomposition and Lagrangian relaxation. The general idea behind both is to remove the linking constraints from the original problem to observe a relaxed problem that decomposes into sub-problems. In the case of Dantzig-Wolfe decomposition the objective function of the appropriate master problem consists of a linear combination of solutions of the relaxed problem. Starting from an initial feasible solution, the subsequent iterations extend this function if the new solution of the relaxed problem verifiably reduces the objective value (i.e., costs). Accordingly, this process is called column generation since the iterations literally creates also new columns in the master problems' coefficient matrix. Flores-Quiroz et al. [61] use this approach in order to decouple discrete investment decisions from dispatch optimization for a GEP with UC-constraints. Although performance enhancements are examined for realistic applications of different sizes these improvements are only quantified for small model instances due to memory issues of not-decomposed benchmark models (ca. 3 times faster, 95% less memory usage).

### 2.4.2. Lagrangian Relaxation

The Lagrangian relaxation is derived from the common mathematical technique of using Lagrange multipliers to solve constrained OPs where linking constraints are considered in the form of penalty terms in the objective function of the master problem. In the applied literature, Lagrangian relaxation is used by Virmani et al. [62] to treat the linking constraints, that couple individual generation units in the UC problem. More recently, Wang et al. [63] applied Lagrangian relaxation on a security-constrained OPF problem in order to decouple a security constraint that links variables of two scales, contingencies and circuits. However, as the treated problem consists of both linking constraints and linking variables, Benders decomposition is applied additionally.

### 2.4.3. Benders Decomposition

Opposed to the previously described decomposition approaches, Benders decomposition can be applied to OPs with linking variables. The general concept of splitting an OP by this approach is based on fixing the linking variables in the sub-problem(s) using their values from the master problem's solution. To improve the solution of the master, the sub-problems are approximated by additional constraints. These so called Benders cuts in turn rely on the dual variables of the obtained solutions in the sub-problems.

As ESOMs are often formulated as linear programs, due to duality of these problems, a translation of linking constraints into linking variables is possible and thus Benders decomposition can be applied to almost all kinds of ESOMs. Accordingly, it is a frequently exploited decomposition technique in the applied literature. Table 2 lists a number of publications that apply decomposition techniques to ESOMs that are at least partially formulated as linear programs (LPs) or mixed-integer linear programs

(MIP). However, due to the non-linearity of AC-power flow constraints, also non-linear programs (NLPs) are a typical use case considered here.

**Table 2.** Overview decomposition techniques applied to ESOMs.

| Authors | Math. Problem Type | Descriptive Problem Type | Decomposed Model Scale | Decomposition Technique | Decomposition Purpose |
|---|---|---|---|---|---|
| Alguacil and Conejo [64] | MIP/NLP | Plant and grid operation | Time, single sub-problem | Benders decomposition | Decoupling of UC and multi-period DC-OPF * |
| Amjady and Ansari [65] | MIP/NLP | Plant operation | | Benders decomposition | Decoupling of UC and AC-OPF ** |
| Binato et al. [66] | MIP/LP | TEP | | Benders decomposition | Decoupling of discrete investment decisions and DC-OPF |
| Esmaili et al. [67] | NLP/LP | Grid operation | | Benders decomposition | Decoupling of AC-OPF and congestion constraints |
| Flores-Quiroz et al. [61] | MIP/LP | GEP | Time, 1-31 sub-problems, sequentially solved | Dantzig-Wolfe decomposition | Decoupling of discrete investment and UC |
| Habibollahzadeh et al. [68] | MIP/LP | Plant operation | | Benders decomposition | Decoupling of UC and ED |
| Khodaei et al. [69] | MIP/LP | GEP-TEP | Time, two sub-problem types, sequentially solved | Benders decomposition | Decoupling of discrete investments into generation and transmission capacity, security constraints and DC-OPF |
| Martinez-Crespo et al. [70] | MIP/NLP | Plant and grid operation | Time, 24 sub-problems, sequentially solved | Benders decomposition | Decoupling of UC and security constraint AC-OPF |
| Roh and Shahidehpour [71] | MIP/LP | GEP-TEP | Time, up to $10 \times 4$ sub-problems, sequentially solved | Benders decomposition and Lagrangian Relaxation | Decoupling of discrete investments into generation and transmission capacity, security constraints and DC-OPF |
| Virmani et al. [62] | LP/MIP | Plant operation | Technology (generation units), up to 20 sub-problems, sequentially solved | Lagrangian Relaxation | Decoupling of unit specific(UC) and cross-park (ED) constraints |
| Wang et al. [72] | LP/MIP | Plant and grid operation | Space, 26 sub-problems, sequentially solved | Lagrangian Relaxation | Decoupling of DC-OPF and UC |
| Wang et al. [73] | MIP/NLP | Plant and grid operation | Scenarios and time, $10 \times 4$ sub-problems, sequentially solved | Benders decomposition | Decoupling of UC, scenario specific system adequacy constraints and network security constraints |
| Wang et al. [63] | LP | Plant and grid operation | Technology (circuits) and time (contingencies), two sub-problem types, sequentially solved | Lagrangian Relaxation and Benders decomposition | Decoupling of DC-OPF, system risk constraints and network security constraints |

\* Direct Current Optimal Power Flow, \*\* Alternating Current Optimal Power Flow

### 2.4.4. Further Aspects

Besides the already presented decomposition techniques, also further mathematically exact approaches exist that are based on individual information exchange between partial problems. Zhao et al. [74], for instance, use this marginal based approach for independent scheduling in a

multi-area OPF problem. Compared to the heuristics presented above, this can be interpreted as the spatially decomposed counterpart to the (temporally decomposed) rolling horizon approach.

Although decomposition approaches provide the capability to improve the performance of solving independent sub-problems of an ESOM in parallel, these techniques are mostly applied for another purpose which results in the iterative solution of a master and one sub-problem. A complicated mathematical problem, such as a large NLP, is simplified by splitting it up into two problems, a smaller NLP on the one hand and a less complicated problem, such as a MIP, on the other. This applies especially to the examples in Table 2 for which nothing is listed in the column "Decomposed model scale". And even though the most frequently identified, decomposed model scale is found to be the temporal dimension, this usually refers to the separation of sub-annual operation scheduling and long-term investment planning in GEP or TEP. According to Table 2, the other typical application of exact decomposition techniques is decoupling of power-flow or security constraints from an UC model which generally refers to a spatial decomposition.

The computational benefits of parallel computing are especially exploited in the context of stochastic OPs. Here the temporal scale is extended by almost independent branches which are referred to as scenarios. These scenarios represent different possible futures which can be determined in parallel (sub-problems) while the assessment of these several alternatives is done by the master problem. Besides the classical decoupling of investment and operation decisions, this approach is also suitable in the context of short-term scheduling. For example, Papavasiliou et al. [75] apply Lagrangian relaxation to decompose by scenarios for a stochastic unit commitment model with DC power flow constraints. Opposed to most ESOMs, they solve their model on a high performance computer with distributed memory architecture. As is it can be expected, Papavasiliou et al. [75] find a significant speed-up due to parallelization. This performance increase, however, poorly scales with the number of cores (e.g., speed-up factor 7 for a hundred times the number of cores). Nevertheless, the main goal of the presented approach is to stay below a threshold of computing time that is suitable for day-ahead operation planning.

*2.5. Aim and Scope*

Despite the existence of a large number of speed-up approaches for ESOMs, it is not clear which methods are the most promising ones to improve the performance of ESOMs that are used in the field of applied energy system analysis. In addition to the arrow-head structure of the coefficient matrix (presence of linking constraints and linking variables, see Section 1), a majority of these models share three characteristics [27]:

(1) To be able to increase the descriptive complexity of the models, the mathematical complexity is often simplified. This frequently means the formulation of large monolithic linear programs (LPs) which are solved on shared memory machines.
(2) Due to the assessment of high shares of power generation from vRES the time set that represents the sub-annual time horizon shows the largest size (typically 8760 time steps)
(3) A great number of applied ESOMs are based on mathematical programming languages such as GAMS (General Algebraic Modeling System) or AMPL (A Mathematical Programming Language") rather than on classical programming languages. Those languages enable model formulations which are close to the mathematical problem description and take the task of translation into a format that is readable for solver software. For this reason, the execution time of the appropriate ESOMs can by roughly divided into two parts, the compilation and generation of the model structure requested by the solver and the solver time.

For the following analyses, we also use GAMS which is, according to a review conducted by Zerrahn and Schill [27], a very popular modelling language in the field of energy systems analysis. We focus on initially large GAMS models for which total computing time is mainly dominated by solver time.

The general aim of this paper is to systematically assess the effectiveness of different performance enhancement approaches for ESOMs that share the above mentioned characteristics. Rather than the comparison of models that deliver exact the same results, we explore possible improvements in terms of required computing time that can be achieved by implementing different conceptual speed-up techniques into an ESOM while staying within a sufficient accuracy range.

By this means, our aim is not to compare all above presented speed-up approaches, but those which are able to achieve the comprehensibly best performance enhancement. In this context, our hypothesis for the selection of model-based speed-up approaches to be systematically evaluated relies on three basic premises:

(1) We focus on very large LPs that have a sufficiently large size for the computing time to be dominated by the solver time and still maintaining the possibility to be solved on a single shared memory computer. If we implement an approach that allows for reduction or parallelization of the initial ESOM by treating a particular dimension, the highest potential therefore can explored by applying such an approach to the largest dimension. Accordingly:

(2) We emphasize speed-up strategies that treat the temporal scale of an ESOM. A high potential for performance enhancement still lies in parallelization, even though, for this study, it is limited to parallel threads on shared memory architectures. Exact decomposition techniques have the advantage to enable parallel solving of sub-problems. However, we claim that each exact decomposition technique can be replaced by a heuristic where the iterative solution algorithm is terminated early. In this way, the highest possible performance should be explored, because further iterations only improve the model accuracy; however they require more resources in terms of computing time. In addition, according to the literature in Table 2, it can be concluded, that mathematically exact decomposition techniques are applied less often with the objective of parallel model execution, but the separation of a more complicated optimization problem from an easy-to-solve one. For very large LPs this is not necessary. For these reasons:

(3) We only analyze model reduction by aggregation and heuristic decomposition approaches.

## 3. Materials and Methods

### 3.1. Overview

Our evaluation approach should provide an assessment of model-based performance enhancement approaches for a very large ESOM that is intended to produce results for real use-cases. However, this implies a couple of challenges. A proper adaption of a large applied ESOM for the comparison of a broad set of speed-up strategies is very time-consuming. Accordingly, we limit the evaluation to the following performance enhancement approaches:

- model reduction by spatial and temporal aggregation
- rolling horizon
- temporal zooming

Moreover, to meet the requirement for an evaluation of very large ESOM instances, we want to prevent the implementation of speed-up strategies into a model that is easily solvable by a commercial solver. Nevertheless, for having references for benchmarking this must still be possible. Hence, we select an existing ESOM for which we know from experience that obtaining a solution is hard but not impossible.

Besides, for fair benchmarking, it must be ensured that the reference model already performs well, e.g., with regard to solver parameterization. To meet this requirement our first methodological step is to conduct a source code review for the applied ESOM and follow recommendations by GAMS developers and McCarl [76]. Although most of the corresponding hints of the latter aim at the reduction of the GAMS execution time, the main objective of this review step is the identification of source code snippets that cause the creation of redundant constraints. In practical terms, this means an explicit

exclusion of unnecessary cases by broadly applying conditional statements ($-conditions). Otherwise, needlessly large models would be passed to the solver.

Finally, it is essential that all model instances that should be compared are executed on identical hardware which should be exclusively available for the ESOM-related computing processes. Ensuring this across the whole evaluation exercise would require a large number of computers with comparatively large memory (>200 GB) to conduct the analysis within practical time spans. Due to a limited access to such equally equipped computers, we guarantee this only for benchmarks across each particular performance enhancement strategy.

The remainder of this section is structured as follows: The modeling setup consisting of a description of the applied ESOM and its characteristics and data as well as the used solver and its basic parameterization are described in Section 3.2. The implementations of speed-up approaches to be evaluated are then presented in Section 3.3. Finally, we set up an evaluation framework that ensures at least a fair comparison of model performance and accuracy across different parametrizations of a particular speed-up approach.

### 3.2. Modeling Setup

REMix (Renewable Energy Mix for a sustainable energy supply) can also be regarded as a modeling framework since several parameterizations of the REMix model exist which share the same source code but focus on various research questions and thus have different scopes in terms of available technologies, geographical study area and time horizon. The analyses for this study were conducted with two model setups which were partially extended. Although most of our analyses are performed for both of them, the results presented in Section 4 build on the REMix instance presented in [77]. The corresponding LP represents the German power system for an energy scenario of the year 2030. In its basic configuration it is a $CO_2$-emission-constrained DC-OPF problem that considers renewable and fossil power generators, electricity transport within the high voltage transmission grid as well as storage facilities such as pumped hydro power plants and lithium-ion batteries.

In addition, no generation capacities are optimized but capacities of both transmission lines and energy storage are optionally considered for expansion planning. To be able to observe a significant expansion of these technologies, their initial values for installed capacities represent the state of 2015. Hence, the installed capacity of lithium-ion batteries is zero. It needs to be noted that this configuration can lead to loss of load situations if capacity expansion is omitted. This is due to the fact that the power plant portfolio of the underlying scenario relies on the assumption that suitable load balancing capability of the power system can be provided by lithium-ion batteries and additional power transmission capacities.

A fact sheet of the appropriate REMix model setup is shown in Table 3 which also provides information about the input and output data.

**Table 3.** Model fact sheet of the applied configuration of REMix based on [77].

| Model Name | REMix |
|---|---|
| Author (Institution) | German Aerospace Center (DLR), Institute of Engineering Thermodynamics |
| | Linear programing |
| Model type | Minimization of total costs for system operation and expansion |
| | Economic dispatch/optimal dc power flow with expansion of storage and transmission capacities |
| Sectoral focus | Electricity |
| Geographical focus | Germany |
| Spatial resolution | 488 nodes |
| Analyzed year (scenario) | 2030 |
| Temporal resolution | 8760 time steps (hourly) |

**Table 3.** *Cont.*

| Model Name | REMix | | | |
|---|---|---|---|---|
| Input-parameters: | | Temporal | Technical | Spatial |
| | | | Dependencies | |
| | Conversion efficiencies [78] | | x | |
| | Operational costs [78] | | x | |
| | Fuel prices and emission allowances [79] | | x | |
| | Electricity load profiles [80] | x | | x |
| | Capacities of power generation, storage and grid transfer capacities and annual electricity demand [81–83] | | x | x |
| | Renewable energy resources feed-in profiles | x | x | x |
| | Import and export time series for cross-border power flows [84] | x | | x |
| Evaluated output parameters | System costs (objective value) | | | |
| | Generated power | | x | x |
| | Added storage/transmission capacities | | | x |
| | Storage levels | x | x | x |

### 3.2.1. Characteristic Constraints

The majority of the mathematical formulations of REMix is presented in [85]. As discussed in Sections 1.2 and 1.3, the coefficient matrix structure of the corresponding LPs contains linking variables and constraints. Besides variables that are induced by enabling capacity expansion (Equation (3)), a great number of linking elements results from modeling power transmission using the dc approximation (spatially linking) or storage facilities (temporally linking). Furthermore, constraints reflecting normative targets, such as necessary for modeling greenhouse gas mitigation scenarios, cause interdependencies between large sets of variables (spatially and temporally linking). For a better comprehensibility Equations (5) to (8) describe these constraints in a simplified manner, i.e., without conditional statements, additional index sets or scaling factors (as implemented in REMix):

Storage energy balance:

$$\boldsymbol{p}_{s+}(t,n,u_s) - \boldsymbol{p}_{s-}(t,n,u_s) - \boldsymbol{p}_{ls}(t,n,u_s) = \frac{E_s(t,n,u_s) - E_s(t-1,n,u_s)}{\Delta t}$$
$$\forall t \in T; \forall n \in N; \forall u \in U_s; U_s \subset U \tag{5}$$

where $U_s$: set of storage facilities.

DC power flow:

$$\boldsymbol{p}_{im}(t,n) - \boldsymbol{p}_{ex}(t,n) - \boldsymbol{p}_{lt}(t,n) = \sum_{n'} B(n,n') \cdot \boldsymbol{\theta}(n',t)$$
$$\forall t \in T; \forall n \in N \tag{6}$$

$$\boldsymbol{p}_{f+}(t,l) - \boldsymbol{p}_{f-}(t,l) = \sum_{l}\sum_{n} B_{diag}(l,l') \cdot K^T(l,n) \cdot \boldsymbol{\theta}(n,t)$$
$$\forall t \in T; \forall l \in L \tag{7}$$

where: $\boldsymbol{p}_{im}/\boldsymbol{p}_{ex}$: power import/export, $\boldsymbol{p}_{lt}$: transmission losses, $\boldsymbol{p}_{f+}/\boldsymbol{p}_{f-}$: active power flow along/against line direction, $\boldsymbol{\theta}$: voltage angle, $B$: susceptance between regions, $B_{diag}$: diagonal matrix of branch susceptance, $K$: incidence matrix and $L$: set of links (e.g., transmission lines).

Emission cap:

$$\sum_{t}\sum_{n}\sum_{u} \boldsymbol{p}(t,n,u) \cdot \eta_e(u) \leq m \tag{8}$$

where: $\eta_e$: fuel specific emissions and $m$: maximal emissions.

3.2.2. Solver Parametrization and Hardware Environment

In preliminary experiments resulting from a broad spectrum of REMix applications, ranging from country specific cross-sectoral energy systems [86,87] to multi-regional [85,88–90] and spatially highly resolved power systems [77], for monolithic LPs, we observed the best performance in terms of computing time and RAM requirements with the following solver parameters when using CPLEX:

(1)　LP-method: barrier
(2)　Cross-Over: disabled
(3)　Multi-threading: enabled (16 if not otherwise stated)
(4)　Barrier tolerance (barepcomp)

- $1e^{-5}$ spatial aggregation with capacity expansion
- default ($1e^{-8}$): rest

(5)　Automatic passing of the presolved dual LP to the solver (predual): disabled
(6)　Aggressive scaling (scaind): enabled

Especially in the case of the first three solver options, LPs that previously could not be solved within time spans of multiple days, turned out to be solvable in less than 24 h. With regard to the solver parameter 5, the amount of required RAM could be significantly decreased. For example, model instances that showed a peak memory demand of 230 GBs when setting predual to −1, otherwise exceeded the available RAM of 300 GBs. For these reasons, all of the following analyses are conducted with GAMS release 25.1.3 using CPLEX 12.8.0 with the above listed solver parameters. In addition, for all implementations of heuristic decomposition approaches either the GAMS option solvelink = 5 (rolling horizon, temporal zooming) or solvelink = 6 (temporal zooming with grid computing) are used to avoid delay times due to frequent read and write operations on the hard disk.

With regard to available hardware, computers with the following (Table 4) specifications are available:

**Table 4.** Specifications of available computers for solving model instances.

| Processor | Available Threads | Available Memory |
|---|---|---|
| **Dual Intel Xeon Platinum 8168** | 2x 24 @ 2.7 GHz | 192 GB |
| **Intel Xeon Gold 6148** | 2x 40 @ 2.4 GHz | 368 GB |

3.2.3. Original REMix Instances and Their Size

As indicated in Table 3 the applied REMix model performs a DC-OPF which is optionally extendable by capacity expansion planning for storage and transmission infrastructures. Depending on this optional setting, two original model instances can be distinguished referred to as "REMix Dispatch" and "REMix Expansion". Due to the different purposes of the decomposition heuristics to be evaluated, the two original models are only investigated for a sub-set of speed-up approaches. The rolling horizon approach is only sufficiently applicable to dispatch problems since investment decisions for especially short time intervals lead to a significant overestimation of required capacity expansion. In contrast, temporal zooming is explicitly suited for problems that account for capacity expansion.

To get an impression of model size, we measure the number of constraints, variables and non-zero elements of the coefficient matrix reported by the solver after performing the pre-solve routines. The appropriate values are indicated in Table 5. They show that enabling expansion planning is costly, especially with regard to the number of constraints. Compared to the number of variables which is increased by approximately 30%, the number of constraints is more than tripled. Nevertheless, this results in a less dense coefficient matrix since the number of non-zeros is only doubled.

**Table 5.** Characterization of original REMix model instances.

| Original Model Instance Name | Applied Speed-Up Approaches | Number of Variables | Number of Constraints | Number of Non-Zeros |
|---|---|---|---|---|
| REMix Dispatch | • spatial aggregation<br>• temporal aggregation<br>• rolling horizon dispatch | 30,579,396 | 9,214,488 | 69,752,951 |
| REMix Expansion | • spatial aggregation<br>• temporal aggregation<br>• sub-annual temporal zooming | 43,169,135 | 32,805,201 | 137,967,269 |

*3.3. Implementations*

3.3.1. Aggregation Approaches

The implemented aggregation approaches either treat the temporal or spatial scale. In case of the first, simple down-sampling is applied to load and feed-in profiles from vRES. Those parameters are available in form of hourly time series (temporally resolved). For down-sampling they are averaged to achieve a data aggregation and accordingly a reduction of the model size by factor *M*. For instance, when transforming a demand time series and, for reasons of simplicity, index sets of the other dimensions are ignored, the appropriate calculation rule is:

$$d_{\text{agg}}(t_M) \; = \; \sum_t \Pi_t(t_M, t) \cdot d(t)$$
$$\forall t_M \in t_M; M \in \mathbb{N} \tag{9}$$

where: $T_M$: set of merged (down-sampled) time steps, $\Pi_t$: map that assigns time steps to merged time steps and $d_{\text{agg}}$: temporally aggregated power demand time-series.

Setting $M = 4$ thus results in input time series that have a 4-hourly resolution. In other words, instead of $t \; = \; 1, \ldots, 8760$ only $t_M \; = \; 1, \ldots, \frac{8760}{4}$ consecutive data points need to be considered in a REMix instance which we refer to be "temporally aggregated".

With regard to the spatial aggregation methodology, we apply the following data processing: First a network partitioning is performed to define which regions of the original model parameterization are to be merged. Therefore, an agglomerative clustering is used by applying the implementation of this algorithm from scikit learn [91] to the adjacency matrix of the original model's network. We chose this clustering methodology as it ensures that merged regions are only built from neighboring regions. In addition, the clustering algorithm itself scales well with regard to various numbers of clusters.

Secondly, we create network equivalents. The applied data aggregation relies on the premise that regions represent so called "copper plates" which means that transmission constraints are ignored within these areas. As a consequence, most nodal properties, such as installed power generation capacity or expansion potentials as well as power demand are spatially aggregated by simple summation. A special case is the aggregation of feed-in time series. Here a case distinction is applied, where the profiles of renewable power generation are aggregated by weighted averaging. The weights are taken from the installed power generation capacities of the respective regions normalized by the sum over the installed capacities within the aggregated region. If there are no capacities installed (e.g., in the case of green-field expansion planning), the maximal capacities resulting from a renewable energy potential analysis are used.

Data that is related to links, such as power transmission lines, is also specially treated: Transmission lines that would lie within an aggregated region are ignored. The transmission capacities of parallel

cross-border links are summed up, while link lengths that are used for loss approximation and susceptance of parallel lines are combined as it is common for parallel circuits, for instance:

$$B_{\text{agg}}(l_M) \;=\; \frac{1}{\sum_l \Pi_l(l_M, l) \cdot 1/B(l)}$$
$$\forall l_M \in L_M \tag{10}$$

where: $L_M$: set of merged links, $\Pi_l$: map that assigns links to merged links and $B_{\text{agg}}$: susceptance of merged links.

### 3.3.2. Rolling Horizon Dispatch

We implement a rolling horizon dispatch into REMix, a decomposition of the original model in time, where the full time horizon of 8760 time steps is divided into a number of overlapping time periods (intervals). For each of these time intervals only the hourly system operation is optimized. Accordingly, capacity expansion is not considered in the appropriate model instances. This is due to the fact that variables that are related to capacity expansion are not resolved on the temporal scale. These temporally linking elements would prevent an easy decomposition in time and thus limit the application of rolling horizon approaches to dispatch optimization problems.

The emission cap (Equation (8)) is also temporally linking and therefore requires changes compared to the native implementation of REMix. A straightforward approach is the distribution of the annual emission budget to the time intervals. In the simplest case the corresponding distribution factors are constant and calculated from the reciprocal of the number of intervals. More sophisticated distributions may take into account input data such as load and feed-in time series to define sub-annual emission caps that correspond to the residual load. However, such a distribution still does not account for regional differences. For reasons of simplicity we use the constant distribution for our implementation of the rolling horizon dispatch.

Storage facilities are only weakly temporally linking as the appropriate energy balance constraint (Equation (5)) only couples neighboring time steps. The error induced by decomposing in time is small as long as the length of time intervals is much greater than the typical energy-to-power ratio of a particular storage technology. Importantly, the overlap prevents that energy storage facilities are always fully discharged at the end of the evaluated part of a time interval to save costs. In the full time-horizon implementation of REMix this undesired effect is addressed by coupling the very last time step to the initial time step. In other words, it is enforced that the storage levels of the first and the last hour of the year are equal. However, this circular coupling is not suitable concerning the boundaries of sub-annual time intervals.

For the rolling horizon approach this means that full discharging still appears by the end of a computed time interval, but it is weakened the longer the overlap. However, there is a trade-off to be made with regard to the length of overlaps since they imply dispatch optimization of redundant model parts and therefore lead to greater total computing times. Another drawback of using overlaps is also that only sequentially solving of multiple model instances is possible.

The discussed characteristics of the rolling horizon approach require a couple of modifications and extensions of the REMix source code especially with regard to the execution phases. In Figure 3 necessary adaptions are visualized.

(1) A new set $T_i$ that represents the time intervals is defined.
(2) The number of overlapping time steps between two intervals as well as a map that assigns the time steps $t$ to the corresponding intervals (with or without overlap) is defined. With a larger overlap more subsequent time steps are redundantly assigned to both the end of the $i^{th}$ and the beginning of the $(i + 1)^{th}$ interval.
(3) It must be ensured that all time dependent elements (variables and constraints) are declared over the whole set of time steps, whereas their definitions are limited to a subset of time steps that depends on the current time interval.

(4)    A surrounding loop is added that iterates over the time intervals.

(5)    With each iteration a solve statement is executed.

(6)    The values of all time dependent variables are fixed for all time steps of the current interval but not for those that belong to the overlap.

(7)    To easily obtain the objective value of the full-time horizon model, a final solve is executed that considers only cost relevant equations. As all variable levels are already fixed at this stage, this final solve is not costly in terms of performance.

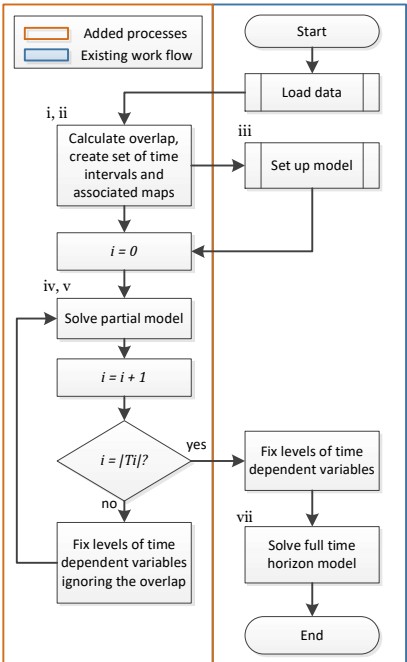

**Figure 3.** Flow chart of implementation of rolling horizon.

The chosen source code adjustments require a manageable amount of effort and can be seen as a processing friendly implementation since all input data is read in the beginning, whereas data is processed slice by slice. Also partial results are held in memory which facilitates an easy creation of a single output file. Established post-processing routines do not have to be changed. Nevertheless, for memory constrained ESOMs, memory friendly implementations are preferable. Data would accordingly be loaded and written to disc slice by slice. The downside of this solution is the fact that these processes must be executed multiple times which results in additional processing costs. Furthermore, the composition of outputs requires a further post-processing that is characterized by multiple read routines of the partial result files.

### 3.3.3. Sub-annual Temporal Zooming

Our implementation of the temporal zooming heuristic is an extension of the previously described rolling horizon approach that enables capacity expansion planning. For this reason, also other temporally linking elements can be treated differently. In particular, each time interval represents a sub-problem where, from a global model perspective, missing information is gathered from a temporally down-sampled full time-horizon model run.

In the case of the storage energy balance, at the boundaries of each time interval the storage level variables are fixed to the levels of the corresponding variables of the down-sampled model's result. Furthermore, for each time interval, factors that define the share of annually allowed emissions are determined with respect to the resulting emissions in the down-sampled model run. This allows a much better distribution of these actually time independent parameter values than an equal distribution as in the implementation of the rolling horizon dispatch.

Even though solving a down-sampled model instance causes additional costs in terms of computing time, the advantage of this approach is the independence of partial models where overlaps are no more necessary. However, as the number of parallel threads is limited on shared memory architectures, this parallelization on the conceptual layer is at the expense of less parallelization on the technical layer, i.e., parallel threads when using the barrier algorithm. For this reason, we implement two versions of the temporal zooming approach (where *I* corresponds to the variable of capacity expansion introduced in Equation (3):

(1) A sequential version that is executed in the same chronological manner as the rolling horizon approach where parallelization only takes place on the solve side (Figure 4).

(2) A parallel version that uses the grid computing facility of GAMS where a defined number of time intervals is solved in parallel. Parallelization takes place on both the model side and the solver side (Figure 5).

Besides the different ways of parallelization the two implementations also differ in the treatment of capacity expansion variables. While in both cases an initial lower bound is defined with regard to the outcome of the down-sampled model run, in the sequential implementation, this lower bound is raised with respect to the results of a particular interval and then shifted to the next interval. On the contrary, the parallel implementation determines the final values of expansion planning variables by selecting the maximum across their interval dependent counterparts.

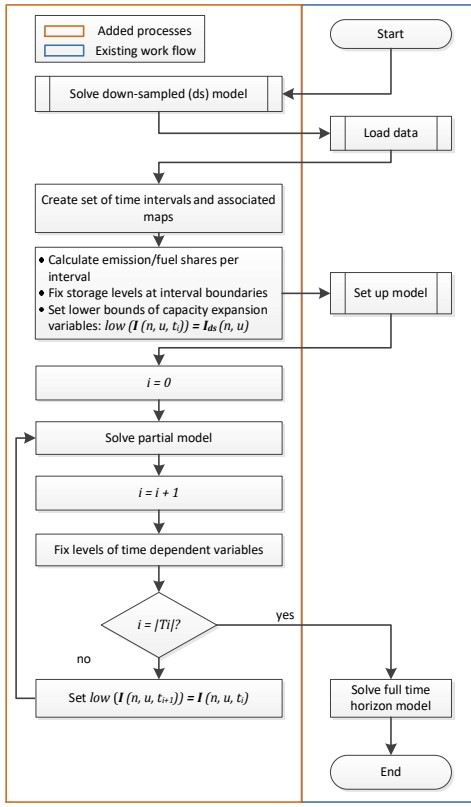

**Figure 4.** Flow chart of sequential implementation of temporal zooming.

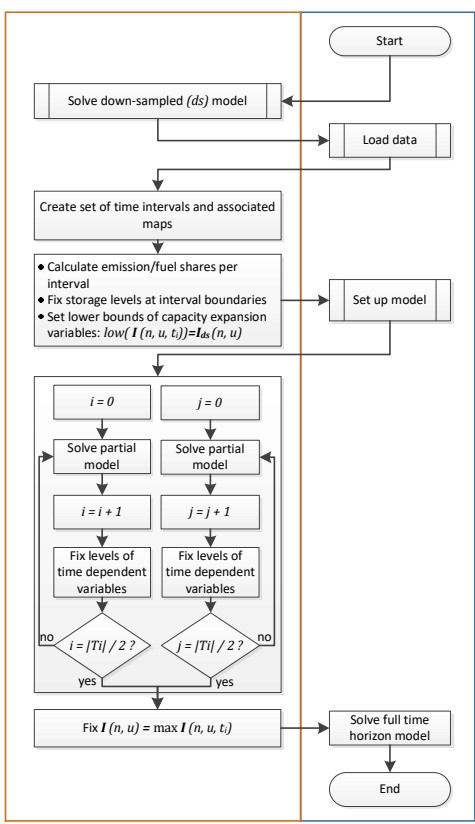

**Figure 5.** Flow chart of grid computing implementation of temporal zooming, exemplarily shown for two parallel runs.

### 3.4. Evaluation Framework

### 3.4.1. Parameterization of Speed-Up Approaches

Each of the implemented model-based speed-up approaches is characterized by parameters that influence the model performance. We refer to these parameters as SAR-parameters (speed-up approach related parameters). In this context, the challenge is to identify SAR-parameter settings that provide both an effective performance enhancement and a sufficient accuracy. We tackle this issue by performing parameter studies. The evaluated parameter value ranges are shown in Table 6.

**Table 6.** Overview of speed-up approach related parameters and value ranges to be evaluated.

| Speed-Up Approach | Parameter | |
|---|---|---|
| | **Name** | **Evaluated Range** |
| **Spatial aggregation** | number of regions (clusters) | {1, 5, 18, 50, 100, 150, 200, 250, 300, 350, 400, 450, 488} |
| **Down-sampling** | temporal resolution | {1, 2, 3, 4, 6, 8, 12, 24, 48, 168, 1095, 4380} |
| **Rolling horizon dispatch** | number of intervals<br>overlap size | {4, 16, 52,365}<br>{1%, 2%, 4%, 10%} |
| **Temporal zooming (sequential)** | number of intervals<br>temporal resolution of down-sampled run | {4, 16, 52}<br>{4, 8, 24} |
| **Temporal zooming (grid computing)** | number of intervals<br>number barrier threads<br>number of parallel runs<br>temporal resolution of down-sampled run | {4, 16, 52}<br>{2, 4, 8, 16}<br>{2, 4, 8, 16}<br>{8, 24} |

In the case of aggregation the SAR-parameters are more or less equivalent to the degree of aggregation. It can be expected that there is a continuous relation between these parameters and the achievable performance and accuracy, where increasing the degree of aggregation will reduce the required computing resources at the expense of less accuracy.

However, the implemented rolling horizon as well as the temporal zooming approaches can be tuned by changing a set of SAR-parameters (Table 6). Thus, the relation between speed-up approach parameterization and the evaluated indicators becomes more complex. For instance, one can expect that there is always an optimal number of intervals with regard to total computing time due to the trade of between faster solving of sub-models and the increasing computational burden from GAMS code compilation.

### 3.4.2. Computational Indicators

When referring to performance we always mean the computing time composed of time spent for model building and solving (solver time). The internal profiling options of GAMS is activated using the command-line option stepsum = 1. All relevant information is then extracted from the logging and listing files of GAMS. The *elapsed seconds* listed in the last step summary represent the total wall-clock time needed for executing all processes. As in our analyses the CPLEX solver is used exclusively, the *solver time* represents the time consumed by CPLEX. This quantity is usually listed above the solver's report summary which also provides the information whether an optimal solution was found. As the CPLEX time reported in seconds can vary depending on the load of the computer system as well as on the used combination of software and hardware, we primarily use the deterministic number of ticks (a computer independent measure) as indicator for required computing time by the solver [92]. The quantity we refer to as GAMS time is accordingly calculated by subtracting solver time from total wall-clock time.

An approximation for peak memory usage is also partially taken from the step summary denoted as Max heap size which represents the memory used by GAMS. An indicator for the memory use on the solver side—in the case of CPLEX's barrier algorithm—is provided by the number of equations and the logging information integer space required [93].

### 3.4.3. Accuracy Indicators

To measure the accuracy of an ESOM one could argue that all variable levels of a model instance treated by a particular speed-up approach should be compared to their counterparts of the original model. However, especially in the case of aggregation approaches the direct counterparts do not always exist. Besides the fact that the computational effort for such a comparison would be great due to the number of variables, an aggregation of the resulting differences would still be necessary to give an indication of accuracy by only a hand-full of comprehensible values. We therefore use only a selection of partially aggregated variable levels for comparison. Nevertheless, we emphasize indicators which are of practical relevance. As indicated in Table 3 these indicators are:

(1) The "objective value" of the optimization problem.
(2) The technology specific, temporally and spatially summed, annual "power supply" of generators, storage and electricity transmission.
(3) The spatially summed values of "added capacity" for storage and electricity transmission, and
(4) The temporally resolved, but spatially summed "storage levels" of certain technologies.

In the following, these indicators are presented relative to the corresponding result of "REMix Dispatch" or "REMix Expansion" observed for conventional solving according to 0. Hence, for example, the accuracy indicator "wind" is determined as follows:

Accuracy indicator "wind":

$$wind = \frac{\sum_{t' \in T, n' \in N, u \in U_w} \boldsymbol{p}'(t', n', u)}{\sum_{t \in T, n \in N, u \in U_w} \boldsymbol{p_{REF}}(t, n, u)} \tag{11}$$

where: $\boldsymbol{p}'$: **variable** levels of total power supply in a model instance treated by a speed-up approach, $\boldsymbol{p_{REF}}$: **variable** levels of total power supply in original modeln instance (without speed-up approach), $T'$: set of time steps in a model instance treated by a speed-up approach, $N'$: set of modeled regions in a model instance treated by a speed-up approach and $U_w$: set of wind enrgy converter technologies.

## 4. Results

### *4.1. Pre-analyses and Qualitative Findings*

#### 4.1.1. Order of Sets

Concerning an efficient execution of GAMS, in addition to the suggestions mentioned in Section 3.1., we observed that it is always advisable to use a consistent order of sets. An illustrative example considering this issue is provided by Ramos in [94]. We also investigated the hypothesis that ordering the index sets from the largest cardinality to the smallest would reduce the time for the model generation. In summary, reductions of up to 40% of the GAMS generation time are observed in some cases. However, the results strongly vary between different model instances. Furthermore, the time spent for model generation can also increase depending on the used version of GAMS. From this experience we conclude that tuning the source code by using particular index orders cannot be considered as a generally effective improvement of model performance.

#### 4.1.2. Sparse vs. Dense

Especially with regard to the way of implementing the equations for storage energy balance and DC power flow, constraint formulations are conceivable that differ from the ones implemented in REMix (Equations (5) to (7)). These formulations make use of fewer variables and constraints and therefore lead to a smaller but denser coefficient matrix. Equations (12) and (13) give an impression of how such dense formulations can look like.

On the one hand, in the case of the storage energy balance equation, the alternative formulation allows that the storage level variables are no more required. On the other hand, instead of an interdependency of consecutive time steps, the power generation or consumption of each time step is linked with all of its previous pendants. This leads to strong linkages across the temporal scale especially for the balance equations that address the elements at the end of the time set. Concerning the DC power flow, Equation (13) can be derived from substitution of the voltage angle and merging of Equations (6) and (7). However, the resulting PTDF matrix requires a matrix inversion that leads to a dense matrix structure:

Storage energy balance:

$$\sum_{t'=t_0}^{t'=t} \boldsymbol{p_{s+}}(t', n, u_s) - \boldsymbol{p_{s-}}(t', n, u_s) - \boldsymbol{p_{ls}}(t', n, u_s) = \boldsymbol{p_{s+}}(t', n, u_s) - \boldsymbol{p_{s-}}(t', n, u_s) \tag{12}$$

$$\forall t \in T; n \in N; \forall u \in U_s; U_s \subset U$$

DC power flow (dense):

$$\boldsymbol{p_{f+}}(t, l) - \boldsymbol{p_{f-}}(t, l) = \sum_n PTDF(l, n) \cdot \left( \boldsymbol{p_{im}}(t, n) - \boldsymbol{p_{ex}}(t, n) - \boldsymbol{p_{lt}}(t, n) \right) \tag{13}$$

$$\forall t \in T; \forall l \in L$$

where: *PTDF*: power transfer distribution factors

The results of our experiments with these alternative model formulations showed that, for REMix, sparse implementations are usually better in terms of model performance. While already small model instances with the dense storage balance equation are nearly unsolvable, the application of PTDF matrices for the DC power flow turns out to be useable but still less performant compared to the implementation that uses the voltage angle.

In this context, on its left y-axis, Figure 6 shows the computing times for two exemplary scenarios (A and B), where, transmission capacity expansion is either enabled or disabled.

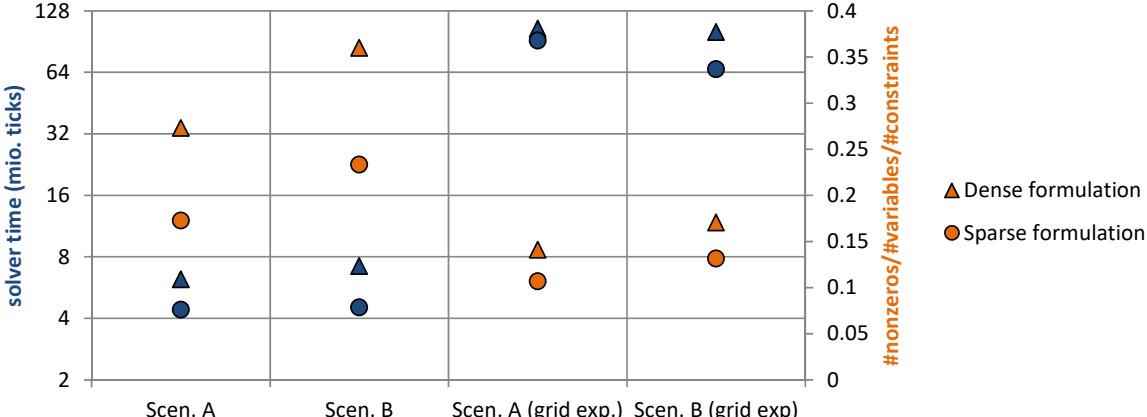

**Figure 6.** Solver time (blue) and non-zero density of the coefficient matrix (orange) for different DC power flow implementations, circles: sparse (with voltage angle), triangles: dense (with Power Transfer Distribution Factors (PTDF)).

The size of underlying model instances ranges between 20 to 38 million variables and 9 to 24 million constraints. To give an indication of the population density of the corresponding coefficient matrices, the number of non-zeros relative to the product of the number of constraints and the number of variables is plotted on the right y-axis. Each of the resulting four model instances is solved using either the dense (triangles) or sparse (circles) DC power flow formulation. As it can be deduced from comparing the blue markers, the computing times for the PTDF-based instances are 15 to 60% greater than in the case of their sparse counterparts. Due to the results of these preliminary experiments the following analyses are exclusively based on model implementations which aim for sparse constraint formulations.

### 4.1.3. Slack Variables and Punishment Costs

A common approach to ensure the feasibility of REMix even for scenarios where the power balance Equation (2) would be violated (e.g., by providing too small power generation potentials) is the use of slack generators. These generators do not have a technological equivalent in the reality and represent the last option to be used in the model for covering a given demand. The associated costs for power supply can be seen as the value of loss of load and thus are high compared to costs caused by real technologies. However, even if very high cost values could be particularly justified by macroeconomic damage, from a model performance perspective it is advisable to set these costs in the same order of magnitude as their real counterparts. Figure 7 shows exemplary computing times of identical model instances of a relatively small size (3 Mio. variables, 2 Mio. constraints). We deliberately analyze small models to prevent the model to run into numerical issues. The differences in the resulting solver time are exclusively caused by changing the model parameter that concerns the costs associated with slack power generation. The increasing computing time with increasing values of this parameter are due to worse model scaling.

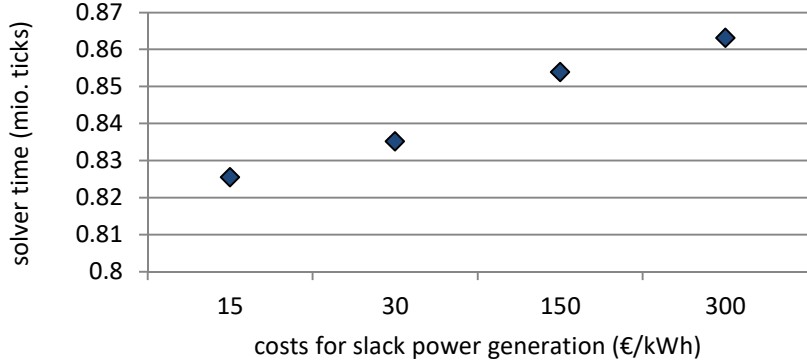

**Figure 7.** Computing time for different values for power generation costs of slack generators.

Despite the fact that scaling is also automatically applied by the solver, it is advisable that in the coefficient matrix of the resulting LP, coefficients stay within a certain range of order of magnitudes. As described by McCarl [95] the factor between the smallest and largest values should ideally be less than 1e5. Since ESOMs such as REMix consider both operational costs of almost zero (e.g., for photovoltaics) and annuities for investments into new infrastructures of several millions (e.g., large thermal units), the corresponding cost ratios are already out of the ideal range. For this reason, the cost factors for slack power generation should not expand this range. Otherwise, especially for large models, the bad scaling leads to numerical issues of the solver and at least extended computing times.

### 4.1.4. Coefficient Scaling and Variable Bounds

Also processing of input data during the generation of equations can pose problems concerning the aforementioned maximum range of coefficients. For example, this is relevant when calculating the fuel consumption based on the power generation divided by the fuel efficiency. Moreover, it is advisable to bound variables to restrict the space of possible solutions which may also lead to a better solver performance. However, finding appropriate bounds for future states of the energy system and claiming to analyze a broad range of conceivable developments implies possible contradictions.

To get a more systematic picture, in Figure 8, we compare a selection of model instances in three spatial resolutions with two different solver precisions. The solver precisions are labelled as "$1e^{-5}$" and "Default" ($1e^{-8}$) while further measures such as explicit rounding of parameters and conscious bounding of variables are varied. The idea behind rounding of input time series and efficiencies is to avoid implicit coefficients with more than five decimals. As a further step in the instance denominated as "bounded variables" we add upper bounds on most variables according to model heuristics. For instance, the power production from slack generators is limited to 10% of the exogenously given electricity demand profile. Additionally, we set upper bounds on decision variables for investments into storage and transmission capacities based on the maximum peak load and annual energy demand of the corresponding regions.

In Figure 8 the conducted comparison is shown for three differently sized instances of both the "REMix Expansion" and the "REMix Dispatch" model. The solver time is depicted relative against the number of ticks required to solve the appropriate model with default settings as presented in 3.2. In this context, the black circles represent the reference values at y = 1.0. While for the small instances with 30 and 120 regions the gains from coefficient rounding (blue markers) seem to indicate better performance, in large scale instances the effect is inverse. For the 488-region instance there is an increase in ticks for the barrier algorithm with the presumably improved numerical properties. In contrast, the additional bounds on variables (orange markers) have a rather little impact on the small-sized instances with only a few regions, while the performance gains for the large scale instances are significant by effectively bringing down the solver time to less than 50% compared to instances with default settings.

From the comparison of triangle and circle markers in Figure 8, it can be furthermore concluded, that the observed effects are independent of the solver precision. However, the possible speed-up highly depends on the general model formulation and may not apply for other solution algorithms than interior point.

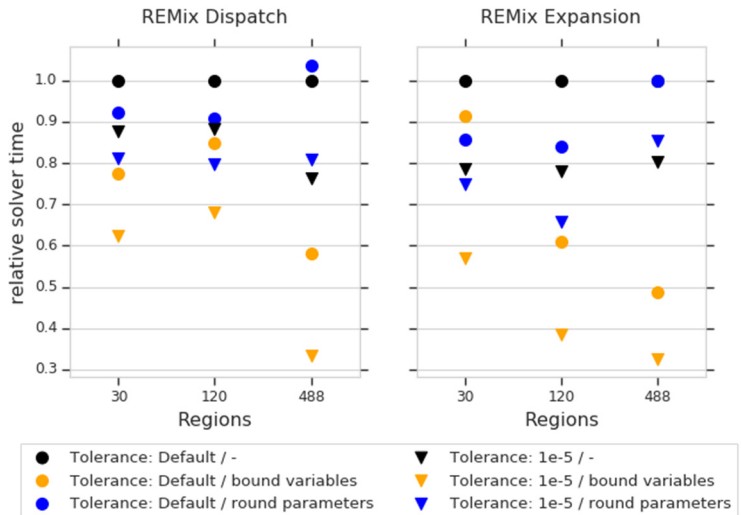

**Figure 8.** Comparison of solver times as a function of numerical properties and solver accuracy.

*4.2. Aggregation of Individual Dimensions*

This section presents the behavior of performance and accuracy indicators for scaling experiments. This means that the original REMix instances ("REMix Dispatch" and "REMix Expansion") are either reduced by spatial or temporal aggregation whereas the degree of aggregation is varied. The number of aggregated regions or time steps of a respective model instance are depicted on the x-axes of the following evaluation figures. In this context, the degree of aggregation is simply defined by:

Degree of aggregation:

$$a(x,v) \; = \; \left(1 - \frac{x(v)}{x_{\text{REF}}(v)}\right) \cdot 100\%$$

$$\forall v \in \{\text{spatial, temporal}\}$$

(14)

where: $x_{\text{REF}}$: $x$-value (number of regions/time steps) of the original model instance.

In the following figures, the curves show computing and accuracy indicators relative to their counterparts of the original model instances. For each indicator, the reference is indicated at the greatest x-value ($x_{\text{REF}}(\text{spatial}) = 488$ regions or $x_{\text{REF}}(\text{temporal}) = 8760$ time steps). Accordingly, the figures are usually read from right to left. The associated absolute y-values are provided in the caption of the respective figure.

4.2.1. Spatial

The results for the spatial aggregation of the "REMix Dispatch" model are shown in Figures 9 and 10. In the former, the computational indicators are depicted by colored curves that represent total wall-clock time, solver time, the number of constraints, the number of non-zeros, and the memory consumed by GAMS as well as an approximation of the memory demand of the solver. On the right hand side, Figure 10 shows the accuracy indicators. Besides the objective value, the annual power generation of selected power generator groups, gas-fired and coal-fired power plants, and wind turbines, is drawn. Even though the REMix model instances consider a broader spectrum of technologies such as photovoltaics, biomass or run-of-river power plants, these technologies are omitted for the sake of clarity.

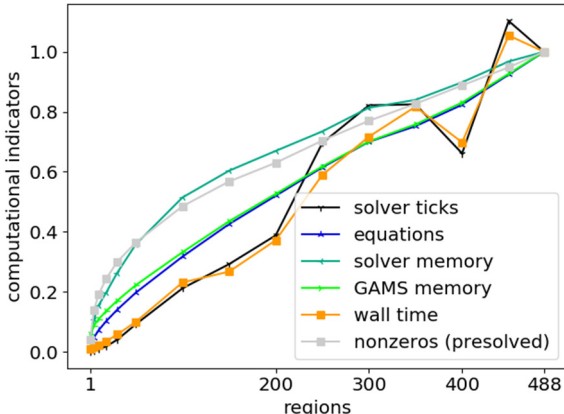

**Figure 9.** Computational indicators for spatial aggregation of the "REMix Dispatch" model. Reference model: CPLEX ticks 16.3 Mio.; Total memory 79 GB; GAMS time 0.6 h; Total wall-clock time 3.6 h.

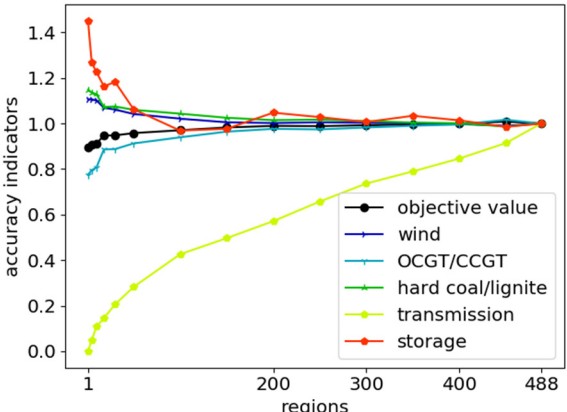

**Figure 10.** Accuracy indicators for spatial aggregation of the "REMix Dispatch" model. Reference model: Objective value 29.7 Bio €; Objective value (cleaned) 21.9 Bio €; Wind 162 TWh; Gas 174 TWh; Coal 105 TWh; Storage 4.1 TWh; Transmission 434 TWh.

With regard to accuracy indicators, up to a degree of aggregation of about 80% (100 regions) most of the curves in Figure 10 show minor deviations within a range of ±5% compared to the reference at $y = 1.0$. While the annual power generation from coal is slightly increasing with stronger aggregation, the opposite can be observed in the case of the objective value and power generation from gas turbines. Wind power and storage utilization are almost constant up to this point. However, for model instances that spatially aggregate to a degree below 100 regions, the use of storage facilities strongly increases. Compared to the reference model, deviations of more than 40% for storage are observable for highly aggregated model instances.

Considering that the number of transmission lines taken into account becomes smaller for more aggregated model instances, it can be expected that most of the effects that come with spatial aggregation stem from unconstrained power transmission. Thus, the strongest influence of this model reduction technique can be observed for the power transmission indicator where deviations greater than 25% already occur for degrees of aggregation >40% (300 regions).

That said, the results can be interpreted as follows: The absence of power flow constraints affects the model accuracy especially when the number of aggregated regions is low and their geographical extent is comparatively large. This facilitates large central power generation units such as pumped hydro storage and coal fired power plants to extensively distribute their electricity in wide areas to the cost of less power generation from probably better sited but more expensive gas turbines.

If the accuracy error for 100 regions is considered to be acceptable for answering a particular research question, the reachable speed-up factor can be determined from Figure 9. For both the solver time (CPLEX ticks) and the total wall-clock time relative to the maximal model time of about 0.2 is observable which corresponds to a speed-up factor of nearly 5. A smaller reduction can be observed for the model size which is characterized by the number of equations as well as the RAM required by the solver (y $\approx$ 0.4) and the GAMS (y $\approx$ 0.3). In terms of reachable speed-up, a linear reduction of the model size by spatial aggregation usually leads to a more than linear reduction of computing time (e.g., solver time), particularly for weak aggregations. However, especially for these model instances a superposed oscillation of the solver time can be observed which makes the estimation of reachable speed-up more uncertain.

For understanding this oscillation better, we analyzed further indicators provided in the logging and listing files as well as more content-related accuracy indicators such as the number of transmission line congestion events or slack power generation. We found that the number of non-zeros appearing within the Choleksy factorization of the barrier algorithm (reported as "total non-zeros in factor") shows a similar behavior. Nevertheless, no correlation between any of the content-related indicators and the solver time was observed. In addition, we cross-checked our results shown in Figures 9 and 10 by performing the scaling experiment with different solver parameters (barrier tolerance $10^{-5}$) as well as based on slightly different clustering algorithm parameters. Both led again to an oscillation of the solver time curve. Thus, we conclude that even if the accuracy indicators scale in a stable manner, especially the solver time depends on how specific nodes are assigned to clusters. Solving of the DC-OPF problem can turn out to be harder for the solver even if the number of regions is smaller than in a less spatially aggregated model instance.

As mentioned in Section 3.2., the initial power plant portfolio of the German power system scenario for the year 2030 is slightly under-dimensioned since storage and power transmission capacities represent the state of the year 2015 ignoring planned expansion of these technologies. In addition, historical weather data of the year 2012 is used which is below the long-time average in terms of renewable power generation. As a consequence the slack power generators are active especially in the "REMix Dispatch" model instances (between 565 and 773 GWh). Total power supply derived from the objective value can thus become more expensive than in the case of "REMix Expansion" depending on the selected specific punishment costs. For this reason, we report two objective values in the caption of the figures of accuracy indictors. Firstly, the objective value of the mathematical optimization problem including costs of punishment terms. Secondly, the cleaned objective value represents costs for total power supply derived from assuming the same costs for slack power generation as for operating fictitious gas turbines.

Figures 11 and 12 show the performance and accuracy indicators for spatial scaling of the "REMix Expansion" model instances. Here, storage (i.e., stationary lithium-ion batteries) and transmission capacities (AC and DC lines) can be added to the system to balance power demand and generation with the installed generation capacities. In accordance to this, the accuracy indicators are extended by storage and transmission expansion. Exceptionally, only the results in this experiment are computed with extensive logging in GAMS's listing files is enabled which automatically leads to an increase of GAMS time.

As reported in the captions of Figures 9 and 11, enabling capacity expansion leads to a significant increase in total computing time from about 3 to almost 50 h. Nevertheless, compared to the "REMix Dispatch" model instances, similarities concerning the over- or underestimation as well as the scaling behavior of the technology specific errors can be observed. For instance, capacity factors of energy storage are increasing for higher degrees of aggregation. This directly affects storage expansion which decreases with the smaller spatial resolution.

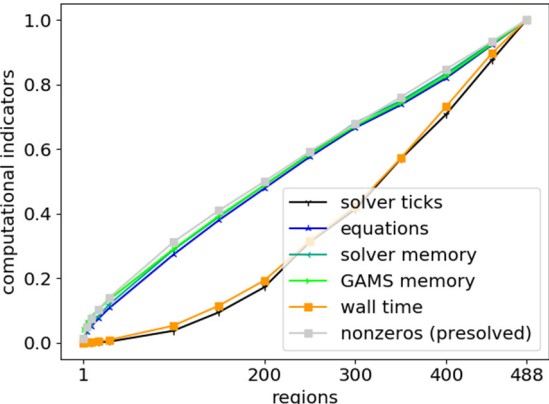

**Figure 11.** Computational indicators for spatial aggregation of the "REMix Expansion" model. Reference model (only in this experiment): CPLEX ticks 381.3 Mio.; Total memory <256 GB; GAMS time 6.6 h; Total computing time 50.9 h.

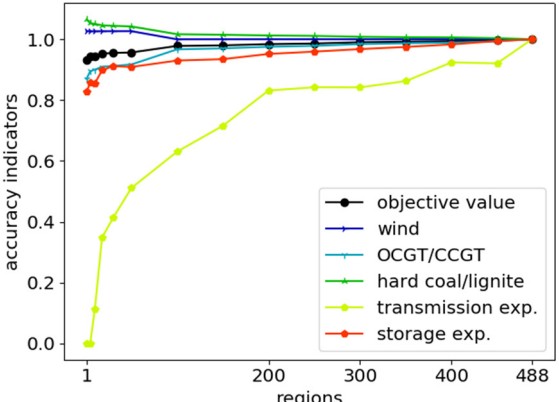

**Figure 12.** Accuracy indicators for spatial aggregation of the "REMix Expansion" model. Reference model (only in this experiment): Objective value 23.7 Bio €; Objective value (cleaned) 23.2 Bio €; Wind 175 TWh; Gas153 TWh; Coal 115 TWh; Storage expansion 123 GWh; Transmission expansion 28.8 GW.

One exception are power transmission-related indicators where more significant deviations from the reference values occur, especially for degrees of aggregation >60% (<200 regions). On the one hand, model instances with such an aggregation even reach reductions in computing time of more than 80%. On the other hand, transmission capacity expansion already experiences significant deviations (>10% compared to the values of the original model) for degrees of aggregation that go below 400 regions. Remarkably, this has only a minor impact on both the objective value and the generation-related accuracy indicators which is observable from the almost horizontal course of the wind, gas, coal, and storage expansion indicators in Figure 12.

A further similarity to the "REMix Dispatch" model is the linear scaling behavior of computational indicators corresponding to the model size as well as the super-linear scaling of the solver time. However, in Figure 11, the solver ticks resemble a rather exponential curve and no superposed oscillation occurs. This means that enabling the expansion of transmission (and storage) capacities leads to a rather expectable scaling behavior of the computing time: The fewer regions in a spatially aggregated model instance, the smaller the time required for solving the optimization problem. If the slope of the solver time curve is regarded as a measure of effectiveness in terms of model acceleration, it can be concluded that spatial aggregation is mainly effective for degrees up to 40%.

### 4.2.2. Temporal

The results for temporal aggregation of the "REMix Dispatch" model are shown in Figures 13 and 14.

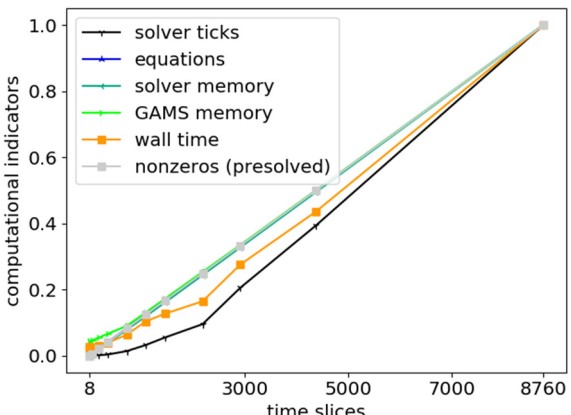

**Figure 13.** Computational indicators for temporal aggregation of the "REMix Dispatch" model.

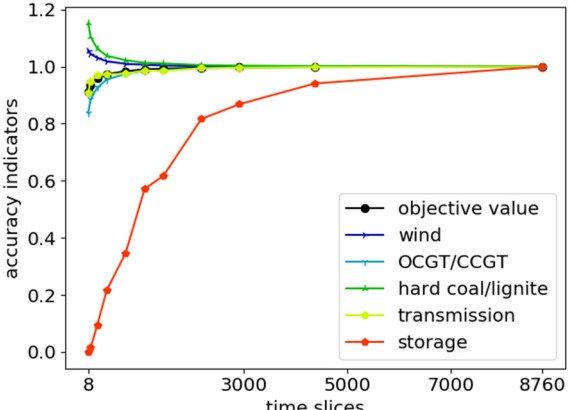

**Figure 14.** Accuracy indicators for temporal aggregation of the "REMix Dispatch" model.

As in the case of spatial aggregation computational indicators are depicted in the figures on the left while accuracy indicators are illustrated on the right. The reference model is the same as in the spatial scenario. In contrast to spatial aggregation, in Figure 14, the slope of the cost curve (objective value) appears much flatter. However, it should be noted that temporal aggregation representing two-hourly time steps already results in an aggregation factor of 50%. For this reason, all of the observed data points in Figures 13 and 14 are located in the half closer to the y-axis. Concerning the solver time this already leads to speed-ups greater than factor 2. Nevertheless, it is not guaranteed that the total computing time (GAMS time + solver time) can be reduced in the same manner. This is due to the additional computing effort for aggregating hourly input data. Compared to such model instances, the greater GAMS time, e.g., in the case of 4380-time steps, results from this additional input data processing. This effect becomes significant for small model instances where the total computing time is not necessarily dominated by solver time. However, for those model instances total computing time is only a few minutes and thus represents no bottleneck. Opposed to this, for the non-aggregated "REMix Dispatch" model the ratio between solver time and GAMS time is still about a factor of 10.

While the objective value as well as most of the technological specific power generation indicators show an absolute error below 5% even for daily averaged time steps (365 time slices; corresponding speed-up factor: 40), significant deviations can be observed for the storage use. For this technology (i.e., pumped storage power plants) the underestimation of power generation compared to the original model is already 5% in the case of diurnal time steps. Also open cycle gas turbines (OCGT) are affected

at degrees of aggregation greater than 70% (e.g., three-hourly time steps). But due to their small electricity production compared to combined cycle gas turbines (CCGT) they have only a minor impact on the slope of the corresponding curve in Figure 14.

Remarkably, power generation from photovoltaics (PV) is almost independent from the degree of temporal aggregation. Because its deviation is less than 0.1‰ across all analyzed model instances, the corresponding curve is not depicted in all figures concerning accuracy indicators. In other words, ignoring day-night periods has no effect on the dispatch of photovoltaics but rather on the need for storage. However, given that in the analyzed model parameterizations the amount of electricity from photovoltaics is only 10% of the annual power generation it becomes clear that PV-integration is possible at almost each point in time. Significant deviations due to temporal aggregation would therefore rather be expected in scenarios with high shares of renewables.

The results for temporal scaling behavior if expansion of storage and transmission capacities is possible can be seen in Figures 15 and 16. For both figures the reference values of the original instance of "REMix Expansion" are denoted a second time. They stay the same for all following analysis with this model.

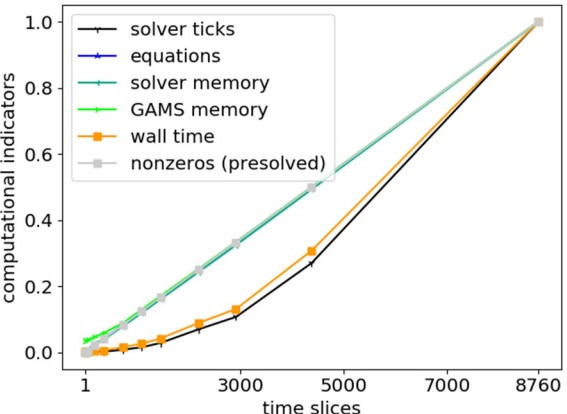

**Figure 15.** Computational indicators for temporal aggregation of the "REMix Expansion" model. Reference model: CPLEX ticks 534.3 Mio.; Total memory >256 GB; GAMS time 0.6 h; Total computing time 62.3 h.

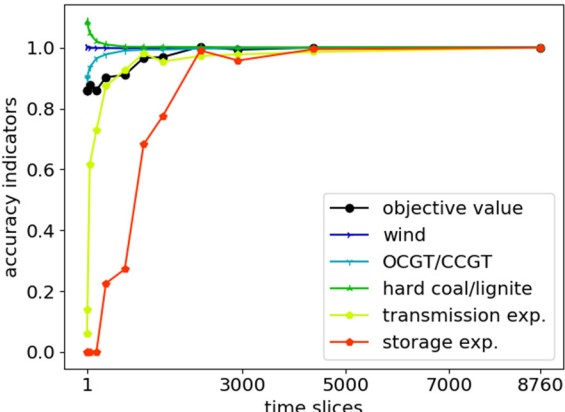

**Figure 16.** Accuracy indicators for temporal aggregation of the "REMix Expansion" model. Reference model: Objective value 22.8 Bio €; Objective value (cleaned) 22.3 Bio €; Wind 180 TWh; Gas 146 TWh; Coal 117 TWh; Storage expansion 122 GWh; Transmission expansion 29.2 GW.

A difference compared to temporal aggregation of the "REMix Dispatch" model instances is the lager area between the green curve that represents the solver time and the blue and violet curves representing the size of a particular model instance. According to this, the reachable speed-up in terms

of solver time is greater for instances with two-hourly (factor 3) or three-hourly (factor 7) time steps. On the other hand, in Figure 15, the slope of the solver ticks is much flatter in its lower part. By this means, going beyond degrees of aggregation of 90% (twelve-hourly time steps) appears to be less effective regarding the reachable speed-up.

Concerning the scaling behavior of model accuracy, significant errors occur for storage-related indicators. Similar to "REMix Dispatch" the annual power generation from storage facilities already decreases by 10% for two-hourly time steps. However, the storage expansion indicator stays below an error of 5% up to an aggregation factor of 75% (four-hourly time steps) while the transmission expansion indicator falls below this value at 730 time slices (twelve-hourly time steps). Therefore, it can be concluded that for observing widely accurate results for capacity expansion of transmission lines and lithium-ion batteries, four-hourly time steps appear to be sufficient, especially assessed against the background of an approximate reduction of computing time by a factor of 13.

### 4.3. Heuristic Decomposition

#### 4.3.1. Rolling Horizon Dispatch?

This section presents the behavior of computational and accuracy indicators for model-based speed-up approaches that make use of heuristic decomposition techniques applied to the temporal scale of both the "REMix Dispatch" and the "REMix Expansion" model. Since the corresponding benchmark experiments vary over different parameters the appropriate figures are built up on hierarchical indices on the x-axes. However, the relative deviations are depicted for each of the analyzed indicators compared to the monolithically solved instances of "REMix Dispatch" and "REMix Expansion". Rolling horizon dispatch

The "REMix Dispatch" model is executed with the rolling horizon approach presented in Section 3.3.2 while the interval size and the number of intervals are varied. The resulting computational and accuracy indicators are shown in Figures 17 and 18. Both the settings for the overlap size and the number of intervals occur on the x-axis.

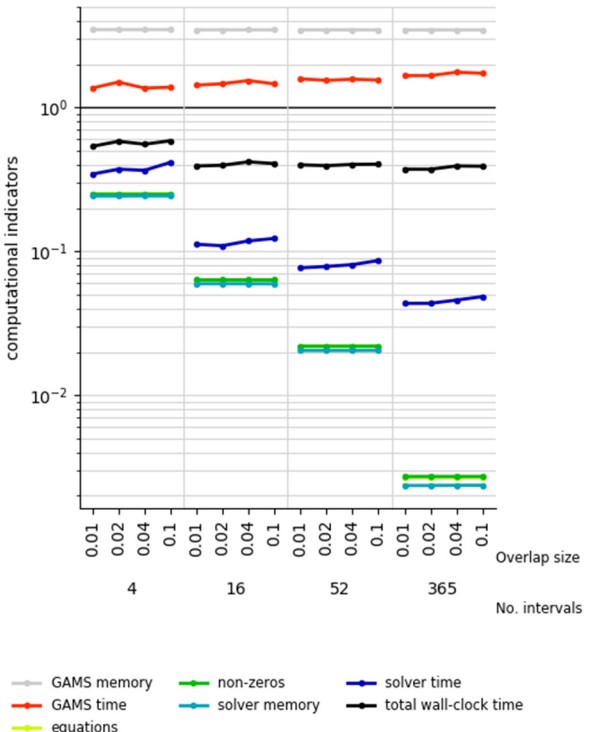

**Figure 17.** Computational indicators for rolling horizon dispatch applied to the "REMix Dispatch" model.

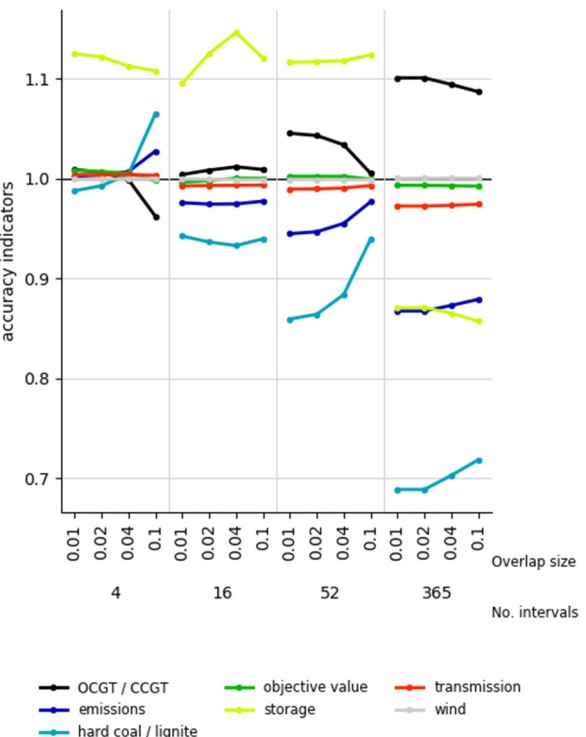

**Figure 18.** Accuracy indicators for rolling horizon dispatch applied to the "REMix Dispatch" model.

With regard to the first, it is striking that the intended behavior of total computing time is achieved—compared to the original model instance speed-up factors between two and three can be observed especially for model instances that decompose the temporal scale into more than four intervals.

In particular, with increasing numbers of time intervals the total time consumed by the solver decreases (down to less than 5% of the monolithic model) as well as the maximal memory required by the solver. On the contrary, memory required and time elapsed for executing GAMS increase by factors around 1.6 and 3.5, respectively.

This is due to the additional need for generating smaller but multiple sub-model instances to be solved one after another. Even though the ratio between GAMS time and solver time is around factor four in the original model instance, when the rolling horizon approach is used, the GAMS time already dominates all model instances but those with four intervals. The total wall-clock time accordingly barely scales with the number of intervals, especially for those with more than 16 intervals.

The overlap size is determined relative to the absolute length of a particular time interval. Compared to the number of intervals, it has only a minor impact on the computational indicators: As it can be expected, the greater the overlap, the more computing resources are required. This is due to the fact that all model parts that lie within the overlap are redundantly considered and thus, the total amount of equations to be solved as well as the number of non-zeros (and variables) increases for greater overlap sizes. However, even if these model size measures increase by 10% (overlap size: 0.1), the resulting total wall-clock time only experiences changes within a range of 2% (4 intervals) to 5% (365 intervals).

Different observations can be made for the accuracy indicators where comparatively large overlaps mostly improve the accuracy of the corresponding model instances. The objective values as well as the indicators for power transport and electricity production by wind turbines have errors smaller than 3% across all investigated model instances. In this context it needs to be considered that we do not observe lower total costs than for the original model instance. Objective values smaller than 1.0 occur since slack generator costs are not considered.

The dispatch of fossil fired power plants and pumped hydro storage units shows stronger deviations. Remarkably for the latter, first overestimations of around 10% are observable for intervals numbers of four, 16 and 52. However, for intervals on a daily level, the storage accuracy indicator shows an underestimation of more than 10%.

These deviations occur, on the one hand, due to the missing circular restriction for the storage level balance that is omitted when the rolling horizon approach is applied. The appropriate constraint enforces the equality of storage levels at the beginning and at the end of the analyzed time period and thus prevents a total discharge for monolithic model instances with perfect foresight. Opposed to that, without this constraint and due to the limited foresight, (even for large overlap sizes in model instances with rolling time horizons) storage levels still tend to zero at the end of an interval ("discharge effect") and thus, average storage levels are smaller than when comparatively long time spans are considered. For example, the mean storage level of 4.6 GWh in the model instance with 365 intervals and 10% overlap is significantly smaller than in the case of four intervals with the same overlap size (20.7 GWh).

In particular, when time interval lengths are in the range of typical storage cycling periods (in the presented case daily periods for pumped hydro storage), storage charging over several energy surplus periods is not cost-efficient for an individual time interval and, in addition, the overlap size cannot be large enough to compensate the "discharge effect". Such a tipping point can be seen in Figure 18 for the 16-interval model instances where storage utilization first increases but decreases as soon as the overlap size changes from 4% overlap (21 h) to 10% (55 h).

On the other hand, the overutilization of energy storage in model instances with less than 365 time intervals stems from another effect. As shown in the upper part of Figure 19, significant deviations between the storage levels of the original (solid black line) and the model instance with seasonal rolling horizon time intervals (solid green line) occur mainly in the middle of the observed scenario year. Furthermore, in the case of weekly intervals (solid grey line), differences from the shape of the black curve appear over the whole time period.

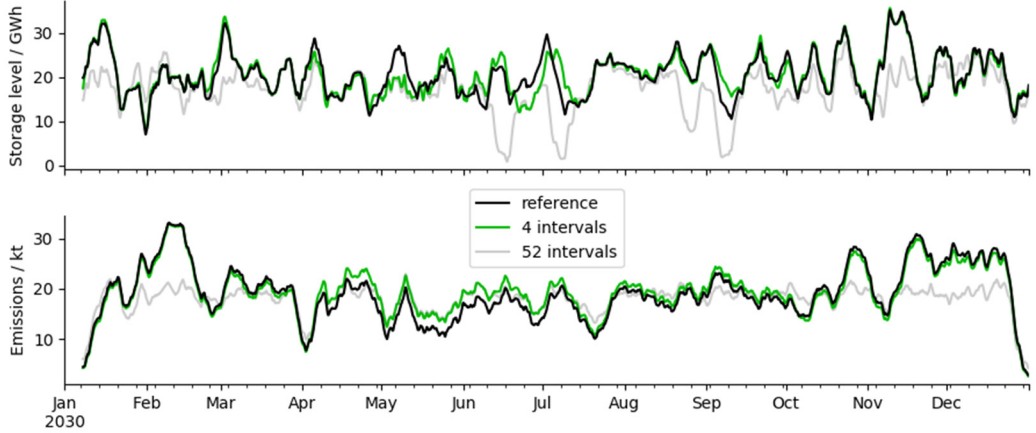

**Figure 19.** Weekly rolling average of spatially cumulated storage levels (top) and greenhouse gas emission (bottom) for two model instances with four and 52 time intervals, computed with the rolling horizon approach, compared to the corresponding results of the original "REMix Dispatch" model instance (reference).

The deviations in storage dispatch occur independently of the intersection areas of time intervals. The reason for this is related to the treatment of the annual greenhouse gas emission budget. In the current rolling horizon implementation the annual emission budget is simply equally distributed to the individual time intervals:

Proportional emission budgets:

$$m_i(i) \;=\; \frac{m}{|T_i| + |T_0(i)|}$$
$$\forall i \in \{T_i\} \tag{15}$$

where: $t_0$: set of time steps that belong to overlaps

According to Equation (15), the resulting cumulated proportional emission budget can be greater than its annual counterpart. However, this especially applies when the absolute size of overlaps becomes large. The reason therefore is the following: Although emission produced within the overlaps are not considered for the final result, model setups exist where the proportional emission budget (that considers also emissions for the time steps within the overlap) is almost fully utilized within the time steps before the overlap begins and thus the total emission may be higher than intended. In Figure 18 this can be observed for the model instance with 4 intervals and 10% overlap. With regard to emissions we call this "negative overlap effect" in the following.

Apart from that, the equal distribution of allowed greenhouse gas emissions rather leads to less total emissions than in the original model instance as they are caused by fossil-fired power plants which are usually in operation in time periods with less electricity feed-in from renewable energies. Such time periods with high residual load are naturally not equally distributed. Consequently, according to the blue lines in Figure 18 and the grey line in the lower part of Figure 19, the more time intervals are considered the more restrictive the proportional emission budget. This also leads to the decrease in dispatch of coal-fired power plants observable for an increasing number of intervals in Figure 18.

Moreover, also the over-utilization of energy storage can be traced back to this effect: In the case of seasonal time intervals, in time spans with low residual load, the slightly higher emission potential allows a technology shift from flexible gas-fired turbines to less cost-intensive coal-fired power plants where the missing flexibility of that latter is provided by energy storage facilities ("negative interval effect"). This finally results in the deviating storage levels and higher emissions for the seasonally sliced model instance in Figure 19 observable in the middle of the analyzed scenario year. The opposite of this technology shift takes place when the emission limit is binding for time periods with high residual load ("positive interval effect"). In this case emission-intensive power generation of coal-fired power plants needs to be replaced by electricity production based on gas. Energy storage then comes into play to increase the capacity factor of CCGT and OCGT plants. However, as it can be seen especially for weekly time intervals in Figure 18, this "positive interval effect" is compensated by the "negative overlap effect".

### 4.3.2. Temporal Zooming

This subsection presents the results for the sequential implementation of the temporal zooming approach applied to "REMix Expansion" model. In this regard, sequential means that multi-threading is only used on the solver level. For a better understanding, we refer to the execution of the temporally down-sampled model instance as "first execution phase" while post-sequent solving of multiple temporally decomposed models is denoted as "second execution phase". In Figures 20 and 21 the resulting performance and accuracy are shown where the parameterization of these two execution phases (temporal resolution of the down-sampled model instance and the number of intervals) is varied. As for the visualization of computational indicators in case of the rolling horizon approach, the x-axes in Figure 20 are hierarchically labeled for the variation of two SAR-parameters (see Section 3.4.1). In this figure, computing times represent cumulative quantities while for the GAMS memory the maximum value is shown. Opposed to that, the indicators that concern the number of non-zeros, the number of equations and the memory demand by the solver show average values reported when solving each sub-model.

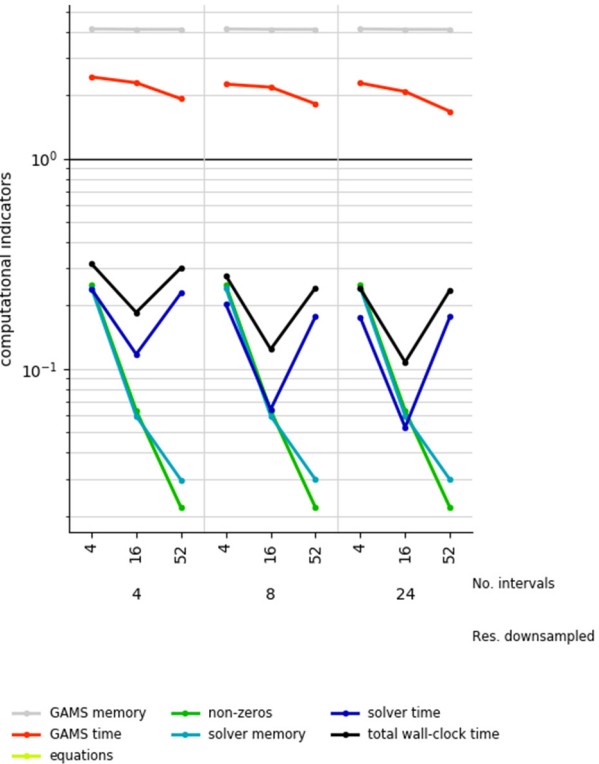

**Figure 20.** Computational indicators for sequential temporal zooming applied to the "REMix Expansion" model.

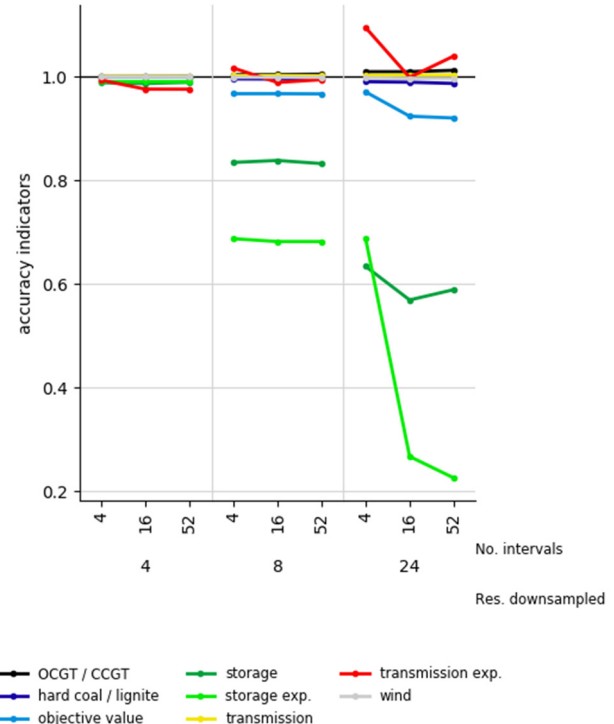

**Figure 21.** Accuracy indicators for sequential temporal zooming applied to the "REMix Expansion" model.

Given that all computational indicators scale with temporal aggregation (see Section 4.2.2), it can be expected that the stronger the temporal aggregation of the down-sampled model instance, the less memory and computing time is required. This expectation matches the results shown in Figure 20.

Furthermore, obvious similarities compared to the computational behavior of the rolling horizon dispatch (see Section 4.3.1) can be observed for the GAMS related indicators. Both the GAMS time and the required memory significantly increase compared to the monolithic reference model. Nevertheless, opposed to the observations made for rolling horizon, GAMS execution times are slightly reduced for an increasing number of time intervals. The total wall-clock time, however, is significantly dominated by the solver performance as the ratio between solver time and GAMS time is greater than factor 100 for the original model and never below 1 for the model instances computed with temporal zooming. Therefore, in Figure 20, the shape of the black curve mirrors the shape of the dark-blue curve that depicts the solver time.

Concerning the solver time, it is striking that there is a significant minimum observable for 16 intervals. This means, even though the solver time can be reduced due to creation of smaller partial models for shorter time intervals, a tipping point exists, when this reduction cannot anymore compensate the additional computing effort for solving multiple sub-models. It becomes clearer when the super-linear scaling behavior for model instances with different numbers of time steps is taken into account. As discussed for Figure 15 in Section 4.2.2, the slope of the curve that represents the scaling of solver time vs. model size, is much flatter for small models (between one and 168 aggregated time steps) than for large models (between 1095 and 8760 time steps). In a temporally decomposed model with four time intervals, the length of an individual interval lies at 2190 time steps and therefore, a more than linear reduction of solver time can be expected. Opposed to that, for 52 time intervals, the time span that is covered by a single sub-model is 168 time steps. In this area of the scaling curve in Figure 15, a reduction of model size by factor two only causes a reduction of total computing time of less than 0.1%.

This decreasing effectiveness of model reduction is also the reason for the less significant increase of speed-up when comparing the total wall-clock time for different temporal resolutions in the "first execution phase". Although the model size between the instances with an eight-hourly and a 24-hourly down-sampled basis is reduced by factor three, the reduction in total computing time is around 1–3%. In contrast, when the instances with 4-hourly and 8-hourly down-sampled bases are compared, the model size is only halved, while the total wall-clock time shows a reduction of 2–6%.

In summary, it can be concluded that speed-ups around factor eight to nine can be achieved. However it needs to be considered that, due to the super-linear scaling behavior, saturation takes place in terms of further performance enhancements.

The error of accuracy indicators of the model instances that are treated by the temporal zooming approach is especially small if a temporally down-sampled model instance with four-hourly resolution is used. It stays below 3% for all accuracy indicators whereas, compared to the outcome of the original model, the largest deviation is observable for transmission expansion when more than seasonal time intervals are considered. For stronger temporal aggregations in the "first execution phase", significant underestimations of storage expansion as well as of storage utilization occur in Figure 21. However, while in case of an eight-hourly resolution the impact of different interval sizes is rather negligible, down-sampling on daily level results in large errors across interval sizes especially for storage expansion.

Given that the storage capacity expansion concerns lithium-ion-batteries that are usually used to smooth the daily feed-in pattern of PV plants, it becomes clear that those energy storage facilities are no longer necessary in the 24-hourly down-sampled model instance. The sudden decrease of the storage expansion for greater numbers of intervals can be accordingly explained as follows:

As for the "second execution phase" lower bounds for investments into new capacities are taken from the results of the "first execution phase", this lower bound is obviously binding for models based on the eight-hourly down-sampled model instance, regardless of the number of intervals in the "second execution phase". For this reason, the storage expansion indicator is at approximately $y = 0.7$ (light-green line). Opposed to that, in the 24-hourly case (right section of Figure 21), the lower bound gathered from the "first execution phase" is considerably smaller as it is depicted in the case

of weekly time intervals ($y = 0.22$). However, additional storage expansion appears particularly for seasonal time intervals ($y = 0.69$). It can therefore be concluded that the shorter the observed time periods of a sub-model, the less attractive are investments into storage capacities.

The objective value accordingly decreases the less storage capacities are built. In this context, it is necessary to have in mind that the effective objective value still includes additional costs for slack power generation and, opposed to the cleaned costs in Figure 21, total costs for power supply are not automatically lower than in the original model.

### 4.3.3. Temporal Zooming with Grid Computing

When we apply the GAMS grid computing facility to the temporal zooming approach, an additional SAR-parameter is to be considered. Although the total number of parallel threads is limited by the available processors on a shared memory machine (in the current study we use 16 threads), their utilization is variable in the grid computing case. While in the previous analyses all 16 threads are used for parallelization of the barrier algorithm, in this section, also the capability to run several GAMS models in parallel is examined. Therefore, the variation parameter "Threads", indicated on the x-axes of Figures 22 and 23, distinguishes the number of runs times the number of parallel barrier threads accessible for the solver.

Opposed to the sequential implementation of temporal zooming, we do not show results for a variation of the temporal resolution used in the "first execution phase" but only for model runs based on an eight-hourly down-sampled instance. This is due to the fact that for the relation between this SAR-parameter and accuracy, it can be expected that the findings from Section 4.3.2 also hold true for benchmark experiments with temporal zooming and grid computing. Using a down-sampled model instance with eight-hourly resolution represents a compromise between desired high speed-up and acceptable loss in accuracy.

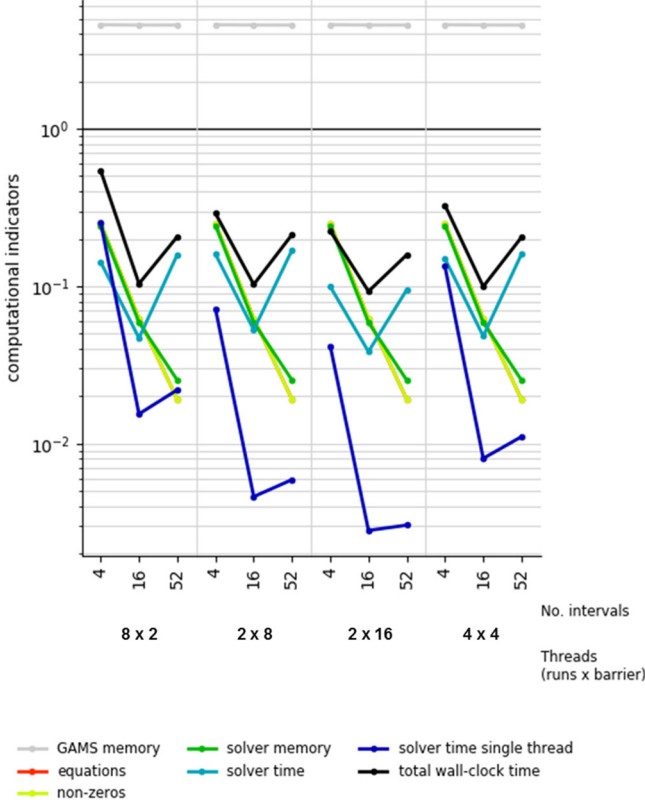

**Figure 22.** Computational indicators for temporal zooming with grid computing and eight-hourly down-sampled basis applied to the "REMix Expansion" model.

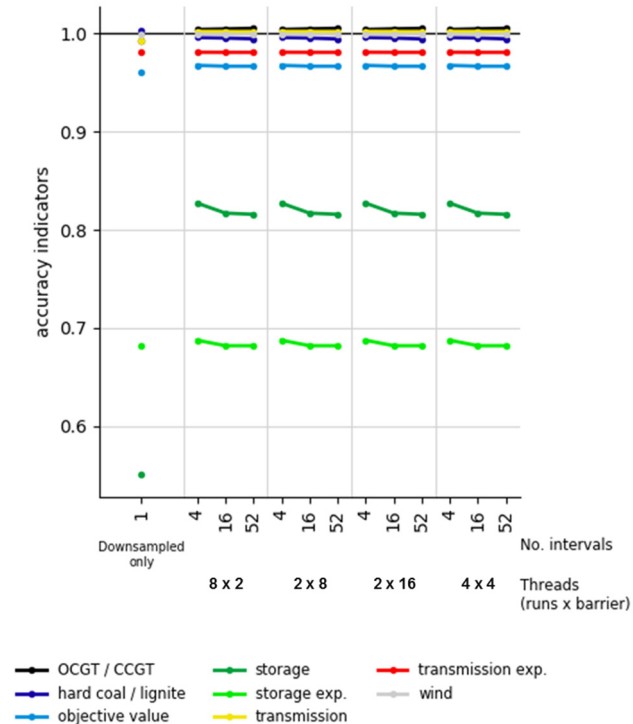

**Figure 23.** Accuracy indicators for temporal zooming with grid computing and eight-hourly down-sampled basis applied to the "REMix Expansion" model.

Furthermore, for efficient in-memory communication between GAMS and the solver the current analysis is conducted with the GAMS option solvelink = 6. This implies that the sub-models that represent the different time intervals are solved in parallel in an asynchronous manner while partial results are hold in memory.

Depending on the combined settings of the number of intervals and the number of parallel threads, the majority of model instances cannot completely be solved in parallel. For example, in the case of 16 intervals and eight threads (and presuming almost equal solver times) it is likely that two sets of sub-models are treated after each other. First, time interval one to eight is solved within eight parallel threads and afterwards time interval nine to 16. In the following we refer to this as "serial part". However, due to the asynchronous solution process and non-equal solver times, for the described example, it is not guaranteed that each thread processes exactly two sub-models.

Given that the machine independent, total solver time (reported in ticks) is not provided by the GAMS logging files, but for each time interval, we post-process the solver time indicator for the performance evaluation. For this reason, solver time is depicted in two forms in Figure 22: The dark blue line, denoted as "solver time single thread", represents the median calculated over the solver times of all time interval-specific sub-models. To account for the "serial part" we multiply this indicator by a factor $\alpha$ to determine an approximation for the effective "solver time" (light-blue line):

Serial solve factor:

$$\alpha = \frac{|T_i|}{n_g} \tag{16}$$

where: $n_g$: number of threads for parallel runs when using grid computing.

In this context, a clear distinction between solver time and GAMS time is also difficult since generation (part of the GAMS time) and solving of particular sub-models are executed in parallel. Deriving an approximation for the GAMS time and normalizing it with respect to its counterpart of the original model appears accordingly less useful. The appropriate computational indicator is therefore not depicted in Figure 22.

Looking at the results for the total wall-clock time, a similar relation between computing time and the number of intervals can be observed as for sequential temporal zooming. Independent of the settings regarding the distribution of threads, the best performance occurs for 16 intervals. On the one hand, this is due to the decreasing effectiveness of model reduction as explained in Section 4.3.2. On the other hand, considering the number of parallel runs $n_g = \{2, 4, 8\}$, it becomes clear, that especially instances that are decomposed into a number of intervals that represents an integer multiple of $n_g$ are candidates for high speed-ups. In these cases the available resources (threads) can be equally utilized. This applies to all model instances with 16 time intervals but only occasionally for seasonally and weekly decomposed model instances.

The most important outcome shown in Figure 22 is the achievable speed-up compared to the sequential temporal-zooming approach. For 16 time intervals and 4 x 4 threads the resulting total wall-clock times go down to values of 10% of computing time of the original model. This additional speed-up appears due to the following effects: In contrast to a pure parallelization on the solver level, grid computing also allows to execute the model generation at least partially in parallel. Furthermore, it can be shown that computing times for implementations of the barrier algorithm in commercial solvers often scale only up to 16 parallel threads [96]. A further reduction of computing time by stronger parallelization (>16 threads) is accordingly only beneficial if it is applied elsewhere within the computing process. Logically, the application of grid computing is especially useful, if more than 16 threads are available in total.

However, the current benchmark analysis shows that parallelization by grid computing is similarly effective as solver parallelization for comparably small numbers of threads. As depicted in Figure 22, different distributions of the number of parallel model runs and the number of barrier have a rather small impact on resulting solver and total wall-clock times. Also for more than 16 threads the additional value of grid competing can only poorly be demonstrated: Taking into account the results for the model instance labelled with $2 \times 16$ threads, it can be stated that despite the total number of threads is doubled, only slight improvements concerning the computing speed are achieved (speed-up factor <10.8).

Apart from that, Figure 23 shows the accuracy for temporal zooming with grid computing relative to the original model instance but also against the outcome of the eight hourly down-sampled model instance used computed in the "first execution phase". For storage utilization significant improvements are observable: While in the down-sampled model instance the accuracy is only 55%, it reaches levels around 82%. This increase in accuracy, however, comes with the costs of less performance (for pure down-sampling on an eight-hourly basis the speed-up is around factor 37). Nevertheless, as discussed in Section 4.2.2, the strongest errors occur with regard to storage utilization and storage capacity expansion. Other accuracy indicators (e.g., transmission expansion) deviate less than 6% from the solution of the original model instance. If only dispatch-related indicators, such as capacity factors of wind, gas-fired or coal-fired power plants are assessed, the appropriate error is smaller than 1%. This outcome is only slightly affected when the number of intervals differs. As discussed in Section 4.3.2 for Figure 21, this SAR-parameter only plays a role if the "second execution phase" is based on down-sampled model instances that show stronger temporal aggregations than eight hourly time steps.

*4.4. Temporal Aggregation Using Feed-in Time Series Based on Multiple Weather Years*

This section exemplary shows the response of accuracy indicators against a variation of model input parameters. Rather than a systematic sensitivity analysis of a broad spectrum of parameters and assumptions across all analyzed speed-up approaches, it emphasizes one particular quantity that is associated to high uncertainties for energy scenarios—the availability of power generation from vRES. The appropriate model parameters to be varied are the hourly feed-in time series for electricity generation from wind and solar energy. While in the analyses above this parameter set is always based on weather data of the year 2012, in the following results for additional weather data of the years 2006 to 2010 are shown.

In Figure 24, the accuracy indicators for temporal aggregation (0) applied to "REMix Dispatch" are depicted. Again, each point represents the relative deviation of an aggregated model's result compared to its counterpart of the non-aggregated model. Therefore, for each weather year an individual original model instance is required and computed (and thus all curves in Figure 24 share the point at time slices = 8760 and accuracy = 1.0).

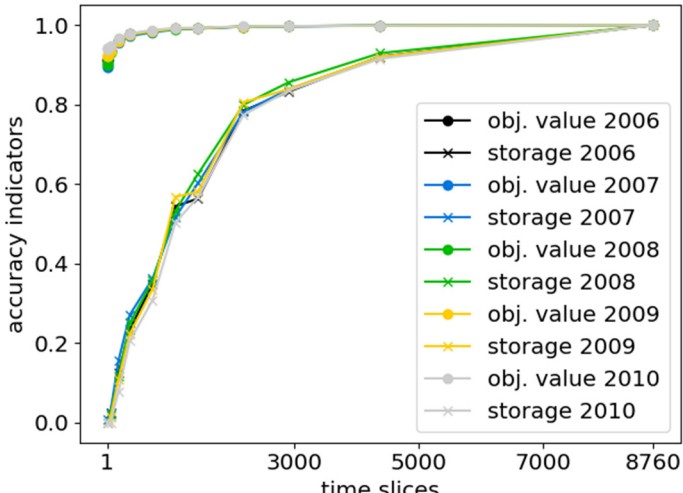

**Figure 24.** Accuracy indicators for spatial aggregation of the "REMix Dispatch" model for parameter variation concerning power generation from wind and solar energy (weather data from 2006 to 2010).

For the sake of clarity only the curves of the objective value and the annual power consumption of energy storage are shown. Nevertheless, considering the findings from above it can be concluded that also the remaining accuracy indicators would show curves that are very similar if weather years are varied. This even applies to the computational indicators which are not significantly affected by the variation of this parameter sets.

As the results in Figure 24 proof, our findings on the impact of the degree of aggregation on accuracy are robust against a parameter variation of weather years. In the first place, this particularly holds true for the case of temporal aggregation applied to "REMix Dispatch". However, it can be expected that even for spatial aggregation and also in the case of "REMix Expansion" deviations of accuracy indicators across different feed-in time series are comparably small. For more general statements concerning sensitivity of accuracy deviations, however, more extensive parameter variations are required where also different assumptions on cost parameters are considered.

## 5. Discussion

### 5.1. Summary

With this paper, we provide systematic evaluations of different approaches to improve the computing performance of applied ESOMs. Besides a number of preliminary measures such as source code reviewing and solver parameterization based on experiences gathered from former model applications, we implemented two kinds of commonly used speed-up approaches to the ESOM REMix. These are, on the one hand, spatial and temporal aggregation methods that showed effective speed-ups up to factor 10 if expansion of storage and transmission capacities is to be considered.

We showed that the majority of analyzed accuracy indicators stay within an error range of about 5% reaching computing time reductions of 60–90% for spatial and temporal aggregation, respectively. Moreover, if particularly affected technologies such as either power transmission or storage are of secondary interest, for dispatch models speed-up factors between 4 and 20 are possible. In this context, it is important to select an appropriate aggregation approach based on the model outputs to be

evaluated in particular. For example, if the competition between technologies that provide spatial or temporal flexibility to the energy system is to be examined, the presented aggregation techniques are not suited for this purpose. For model instances that consider capacity expansion, we also observed that significant speed-ups are particularly reached for low to intermediate degrees of aggregation. In contrast, strong aggregations (beyond 90%) showed only relatively small additional improvements in computing performance.

Based on these findings, we conclude that model reduction by aggregation offers the possibility to effectively speeding-up ESOMs by at least factor two without the implication of significant losses in accuracy. In contrast, strong degrees of aggregation are less useful because speed-up gains are comparatively small while accuracy errors reach inacceptable levels ("effectiveness of model reduction").

On the other hand, we applied nested model heuristics that aim at the decomposition of the temporal scale of an ESOM. As these speed-up concepts imply manipulations on the temporal scale of an ESOM, they affect accuracy indicators that are related to modeling energy storage. The benchmark analyses of the rolling horizon approach for pure dispatch-models revealed that large overlap sizes and interval periods that cover full storage cycles are recommendable. Their additional costs with regard to computing effort are low, but may increase accuracy significantly. For the computational performance of the rolling horizon dispatch the ratio between GAMS time and solver time is crucial since only for dominating solver times, significant speed-ups around a factor of 2.5 could be observed for "REMix Dispatch". In this regard, it needs to be considered that "REMix Dispatch" is still a quite easy-to-solve model instance (total wall-clock time <4 h). Based on our knowledge about "effectiveness of model reduction" we assume that this performance enhancement approach will be even faster for larger dispatch models.

Considerably higher speed-ups were observed for the lager "REMix Expansion" model that was treated by the temporal zooming approach. We showed that within the limited capabilities for parallelization on shared memory hardware, speed-ups of more than factor 10 were possible, especially if grid computing was used. However, besides the limitation imposed by hardware resources, the reachable performance enhancement is also restricted due to scaling behavior of very small models. This means, that additionally to the ratio between GAMS time and solver time, it needs to be considered that as soon as sub-models are reduced to a certain size, further size reductions only slightly decrease solver time (downside of "effectiveness of model reduction"). Hence, with regard to speed-up by parallelization, it is remarkable that at first glance, many intervals appear to be more effective. However, according to the results in 0 und 0, medium sized intervals performed best.

## 5.2. Into Context

Our findings, especially concerning temporal aggregation, are also in-line with those of Pfenninger [26] who reports reductions of computing time of more than 80% at three-hourly time resolution for scenarios of the ESOM Calliope applied to scenarios for the UK. With regard to accuracy, Pfenninger reports the values for capacity expansion of wind energy converters. His results show that the higher the wind penetration of a particular scenario is, the stronger the errors that occur due to temporal aggregation. However, the availability of storage technologies puts the effect of strong deviations compared to an hourly-resolved model instance into perspective.

This indicates that the scaling behavior of computing time rather depends on the model characteristics than on the composition of input parameters. Opposed to this, the scaling behavior of accuracy measures indicates a dependency on the parameter setup. However, our exemplary parameter variation across different feed-in time series for one particular use case also indicated certain robustness of the resulting accuracy errors for different degrees of aggregation.

In contrast to the here applied "REMix Expansion" model, Calliope also considers the expansion of generation capacities. In [26], for a scenario with extensive capacity expansion of renewables, the steep decrease of the curve of computing time for low degrees of aggregation is more pronounced

than in our model instances which rather show a smooth transition to the area with a flatter slope ("effectiveness of model reduction").

For the examined heuristic decomposition techniques, our observations concerning accuracy are in-line with expectations derivable from known strengths and weaknesses occurring when differently treating the temporal scale: The down-sampled model instance allows a better approximation of capacity expansion indicators due to the consideration of the full time-horizon to be analyzed. In contrast, solving model instances with the best temporal discretization enables an accurate dispatch of available power generators (and storage units). However, as results for accuracy gains by the latter show, running a temporally decomposed model instance - when the solution for its down-sampled counterpart is known – was only beneficial for observing a more accurate dispatch of storage units or when the temporal resolution in the "first execution phase" was poor. In this case it needs to be considered, that for sufficient accuracy enhancements the selection of an appropriate number of intervals is crucial since errors of accuracy indicators only decrease for comparably large interval sizes.

Given that the "effectiveness of model reduction" becomes more significant when going from the comparatively easy-to-solve "REMix Dispatch" to the more complicated "REMix Expansion" model while it is also observable for different scenarios analyzed by Pfenninger, it can be generally concluded, that already low degrees of aggregation with small accuracy errors become the more valuable the harder it is to solve a particular monolithic ESOM. This makes model speed-up approaches that are based on model reduction techniques even more attractive for application to ESOMs programed with mixed-integer variables.

## 5.3. Limitations

The claim of conducting analyses for comparably large model instances implies several challenges that only partially could be addressed. As mentioned in Section 3.1., the whole benchmarking should ideally be carried out on the same computer hardware ensuring no influence on the solving process by parallel processes of other applications. However, due to a limited access to equally equipped computers, the instances of the „REMix Dispatch" model with rolling horizon were solved on the JUWELS cluster of the Juelich Supercomputing Center (first row in Table 4). For all of the other benchmark experiments other hardware was used (second row in Table 4).

Also minor bug-fixes were applied to REMix between the different benchmark experiments. One remarkable change is the indicated reduction of solver precision from $1e^{-8}$ to $1e^{-5}$ to reduce total computing times for the experiments related to spatial aggregation with capacity expansion (see Section 3.2.2) while extensive logging in GAMS's listing files was enabled. This obviously changed the ratio between GAMS time and solver time and probably led to smaller speed-ups observed for spatial aggregation with instances of "REMix Expansion".

For these reasons, speed-ups found for the individual performance enhancement approaches are not fully comparable with each other. Despite this circumstance, it can be expected that ideal conditions are also hardly achievable if speed-up approaches are used in applied studies. And still, for large models, the relation between achievable speed-up by a particular performance enhancement approach and impact on the computing time by parallel third-party processes should be negligible.

Moreover, the two selected REMix models that were used for this evaluation of speed-up approaches share many similarities with other applied ESOMs, especially if these are formulated in GAMS. However, we do not claim to provide general findings - such as the specific number of intervals to use for a rolling horizon method - that are representative for all of these models. For instance, because our results are only based on a single model parameterization, the impact of different data sets especially on accuracy indicators could not be assessed which limits the general transferability of our findings. Nevertheless, the outcome of this study provides a clear indication which speed-approaches show the highest potential for significantly reducing computing times. Furthermore, we mainly used straight-forward implementations that can still be tuned towards greater accuracy if required. This is

particularly necessary if other indicators than the ones that were used in this study (mainly on an annual basis) are of interest; e.g., shadow prices.

*5.4. Methodological Improvements*

In this paper, we mainly focused on reachable improvements concerning the computational indicators, i.e., the required total wall-clock time. However, as all of the presented methodological approaches do not provide exact solutions of the original model instances, improvements regarding the accuracy can be considered if necessary. In the case of model reduction, a broad variety of conceivable methods to increase the accuracy of particular model outputs exists (see Section 2.2). As methods such as representative time slices or more sophisticated network equivalences are more or less related to smart treatment or preprocessing of input data, the total time consumption for the overall modeling exercise will not significantly increase.

With regard to the applied rolling horizon dispatch approach, similar improvements are conceivable by using temporally aggregated data for the time steps within the overlap. The idea behind is an extension of the foresight horizon while keeping the number of redundant time steps to be considered low. For instance, for the operation of long-term storage, down-sampling of the residual load for the next annual period would be valuable to avoid the undesired effect of full discharging towards the end of an interval.

Moreover, improved estimations for emission budgets for each interval are conceivable. In the actual implementation the annual emission budget is simply equally distributed which, on the one hand, prevents the dispatch of thermal power plants particularly in points in time with high residual load. On the other hand, time intervals where sufficient renewable energy resources are available may require a smaller emission limit instead. To address this issue, it could be considered to shift unused emissions from one time interval to the next and to select a summer date as starting point for an annual model run and heuristic decomposition approaches such as the presented temporal zooming method offer a starting point for improvements that could go into two directions:

(1) Improved performance can be gained by running the independent model parts (such as the time intervals in case of grid computing presented in 0) on different computers. By this means, the drawback of being limited to memory and CPU resources of shared memory machines could be overcome. In this context, for a better coordination and utilization of available computing resources the application of workload managers such as Slurm [97] would be beneficial.

(2) Improved accuracy can be reached by an extension to an exact decomposition approach that decomposes the temporal scale. However, this requires additional source code adaptions. For instance, in case of Benders decomposition, the distribution of emission budgets to the respective intervals needs to be realized by interval specific variables necessary to create benders cuts. Additionally, it can be expected that due to the need of an iterative execution of master and sub-problems the total computing time would significantly increase. Taking into account the best achievable speed-up of 10 of temporal zooming compared to simply solving the monolithic model, there is only a little room for improvements which may be disproportionate to the implantation effort required.

Finally, the combination of both improved performance and maintaining the accuracy requires iterative methods as well as the utilization of distributed memory computing hardware. However, effective implementations of such performance enhancement approaches require efficient communication between the processes that are executed in different computing nodes. Parallelization should therefore not only be thought at the conceptual level but also on the technical layer (see Figure 2). This goes hand in hand with the parallelization of solvers which is realized with the PIPS-IPM++ solver [98]. This solver provides a HPC-compatible implementation of the interior point method for LPs that are characterized by linking variables and linking constraints.

*5.5. Practical Implications*

In this study, the presented use case of a German ESOM has a macroeconomic perspective. It is thus suited for decision support in the field of energy policy of national economies. However, very similarly shaped optimization problems are also to be solved in energy industry.

For example, an aggregator which bundles and markets the power generation of decentralized power plants together with a storage facility across energy markets has to make operating decisions while seeking for a margin maximization that also needs to consider a technological, spatial and temporal component. Although time series in such use cases are not as extensive as in our study, manageable computing times are much more crucial as deadlines for bidding define hard time slots which are available for model-based analyses. In particular, rolling horizon approaches are suitable for dealing with weather and electricity price prognoses errors which become smaller with a shorter foresight horizon of a particular time interval. According to our findings both overlaps and interval sizes need to be comparably large for high accuracy (the latter should be, at least, greater than the typical cycling period of the storage facility). In this context, a high accuracy of the objective value (compared to the global optimum under perfect foresight) is to be understood as the potential to reach a higher margin. Our analysis in 0 shows that implications on total computing time by varying overlap and interval sizes are negligible.

However, this still leaves spaces for further and more detailed case studies because recommendations on the discussed SAR-parameters (0) are only valuable if the concrete framework conditions of an applied use case are known. For example, in our study, nearly constant computing times with rolling horizon dispatch were observed for time ratios (interval size divided by size of total foresight horizon) ranging between factor 0.003 and 0.02. If a total foresight horizon of 48 h would be considered in the example of the aggregator, the appropriate interval sizes would range within less than one hour.

## 6. Conclusions

Energy systems analysis highly depends on modeling tools such as Energy System Optimization models (ESOMs). To fulfill their purpose to provide insights into complex energy systems for decision support they need to be solvable within acceptable time spans.

For the broad spectrum of existing measures to improve the performance of ESOMs, we provided a detailed classification of conceivable approaches. Furthermore, we gave examples on easy-to-use adaptions that already improve computing performance, especially for ESOMs formulated in GAMS. These measures were accompanied by comprehensive benchmark analyses for a set of frequently applied speed-up techniques. The conducted examination included model aggregation approaches on different scales as well as strategies for heuristic decomposition. Both were applied to a spatially (488 regions) and temporally (8760 time steps) highly resolved ESOM of Germany for an energy scenario of the year 2030. While conventional computing with commercial solver software required more than two days for optimal solutions of certain model instances, selected speed-up approaches obtained sufficient solutions after less than six hours.

In particular, the novelty of this paper is the systematic evaluation of a broad set of approaches assessed for an applied ESOM focusing on achievable performance improvements. This allowed statements concerning possible speed-up factors and implied accuracy losses that went far beyond existing, methodologically focused assessments of single approaches with generic model setups.

In this context, Table 7 shows the final overview of the deeply analyzed speed-up approaches of the current study. Here, the "sufficient speed-up" indicates how many times faster a model instance could be solved compared to the total time required to solve the same model in the conventional way. As our analyses emphasized model reduction and heuristic decomposition, "accuracy" was quantified by using a set of pre-defined accuracy indicators. These indicators were determined as the relative deviations of:

- the "objective value" of the optimization problem,
- "power supply" of different electricity generation and load balancing technologies as well as, if appropriate,
- "added capacities" of storage and electricity transmission

against their counterparts calculated with an original, conventionally solved model instance (see also 0). In Table 7, the deviation from 100% accuracy is listed for both, the average over all assessed accuracy indicators and the accuracy indicator that showed the greatest error.

**Table 7.** Overview of analyzed performance enhancement approaches: observed speed-up and accuracy evaluated across all considered accuracy indicators.

| Speed-Up Approach | Sufficient Speed-Up (Model Instance) | Accuracy | |
|---|---|---|---|
| | | Average | Worst (Affected Indicator) |
| **Spatial aggregation** | | | |
| "REMix Dispatch" | >4 (100 regions) | >95% | >70% (power transmission) |
| "REMix Expansion" | >8 (150 regions) | >95% | >70% (transmission expansion) |
| **Down-sampling** | | | |
| "REMix Dispatch" | >6 (2190 time steps) | >97% | >81% (storage utilization) |
| "REMix Expansion" | >10 (2190 time steps) | >97% | >87% (storage utilization) |
| **Rolling horizon dispatch** | ≈2.5 (16 intervals) | >96% | >87% (storage utilization) |
| **Temporal zooming (sequential)** | >8 (1095 time steps/16 intervals) | >93% | >69% (storage expansion) |
| **Temporal zooming (grid computing)** | >10 (1095 time steps/16 intervals) | >92% | >68% (storage expansion) |

According to Table 7, within our evaluation framework, temporal down-sampling turned out to be the most efficient speed-up approach. The usefulness of this approach is strongly related to the "effectiveness of model reduction". In other words, the larger and more difficult to solve a particular ESOM becomes, the greater the achievable speed-up by already minor model reductions is. Taking into account that solving of linear ESOMs with mixed-integer variables is more complicated than for the model instances considered in this study, we suppose that the presented speed-up approaches are especially effective for such use cases.

As far as only specific model outcomes such as additional transmission capacities are of interest and extensive multi-threading is possible, the presented heuristic decomposition approaches with grid computing (temporal zooming) are also promising as they allow additional speed-ups without increasing loss of accuracy. Moreover, they offer the possibility for executing an ESOM on multiple shared memory computers even though parallelization is only applied to the conceptual layer of the optimization model (see Section 2.1).

Nevertheless, we showed that the appropriate gains in performance are limited depending on the size of a certain model. In this case, the down-side of "effectiveness of model reduction" comes into play: Since the idea behind decomposition is based on solving multiple reduced sub-models, such approaches reach their speed-up limit when the decrease of computing time by model reduction becomes negligible for very small sub-models.

Restrictively, the examined speed-up approaches were implemented and evaluated for a single ESOM framework. In this regard, further systematic evaluations are conceivable where variations of both input data and model specific source code need to be done systematically. This especially applies to the latter because we suppose that differing input data affect the accuracy of an ESOM rather than the computing performance.

In conclusion, the capability to solve very large ESOMs much faster is a pre-condition for best-practice studies in the field of energy systems analysis. Rather than spending time on solving models only for a hand full of scenarios and parameter sets, broad parameter scans become possible for which plenty of model solutions are required. In this manner, the application of effective speed-up approaches highly contributes to the generation of robust and well-founded model-based analyses for the development of decarbonization strategies of the energy system.

**Author Contributions:** Conceptualization, K.-K.C. and M.W.; methodology and software, K.-K.C., S.S., M.W. and K.v.K.; formal analysis, K.v.K.; investigation, K.-K.C. and K.v.K.; resources, K.v.K., M.W. and K.-K.C.; data curation, K.v.K. and K.-K.C.; writing—original draft preparation, K.-K.C. and F.C.; writing—review and editing, K.-K.C. and M.W.; visualization, K.v.K. and K.-K.C.; supervision, K.-K.C.; project administration, K.-K.C.; funding acquisition, K.-K.C.

**Funding:** This research is part of the project BEAM-ME. It was funded by the German Federal Ministry for Economic Affairs and Energy under grant number FKZ 03ET4023A.

**Acknowledgments:** We thank our colleagues from the BEAM-ME project who provided insight and expertise that greatly assisted the research. Especially, we would like to thank Fred Fiand and Michael Bussiek from GAMS for their helpful hints and support when analyzing the performance of REMix' source code. We thank Thomas Breuer and Dmitry Khabi for their help how to use the computers at the Supercomputing Centers in Jülich (JSC) and Stuttgart (HLRS), respectively. The authors gratefully acknowledge the Gauss Centre for Supercomputing e.V. (www.gauss-centre.eu) for funding this project by providing computing time through the John von Neumann Institute for Computing (NIC) on the GCS Supercomputer JUWELS at Jülich Supercomputing Centre (JSC). Further thanks go to Daniel Rehfeldt, Ambros Gleixner and Thorsten Koch from Zuse Institute Berlin (ZIB) as well as to Ontje Lünsdorf and Thomas Vogt from the DLR Institute for Networked Energy Systems (DLR-VE) for providing access to computing resources with a sufficient amount of memory. Finally, we would like thank Yvonne Scholz for her comments that greatly improved the manuscript.

**Conflicts of Interest:** The authors declare no conflict of interest. The funders had no role in the design of the study; in the collection, analyses, or interpretation of data; in the writing of the manuscript, or in the decision to publish the results.

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
