# Peer review of "Classification and Evaluation of Concepts for Improving the Performance of Applied Energy System Optimization Models"

_energies, doi:10.3390/en12244656_

Round 1

Reviewer 1 Report

Positives:

1. A very solid and strong overview paper.

2. Very relevant references and techniques accumulated at one place.

3. Very useful for people operating/researching in energy sector. The considered problems are on the spearhead of research for energy market at present.

4. Unique generalization and classification of different "local optimization" problems into design and operating sets.

5. Really important problem of reducing computational complexity is addressed - relevant both for science and business.

Recommendations:

A detailed explanation of a narrowed practical example would be of great use (with all definitions, methods, procedures and results) - this will give obvious "touchable" understanding to the business/industry-oriented user. For example consider a detailed analysis of a partial problem of "maximizing the margin or minimizing fuel consumption for a power plant" (or another one) - maybe this worth writing a separate article on that to show not only academic but also "applied results" A more clear presentation of results comparison is required (for example - see table 1). What digits does the column "accuracy" actually show? 95% of what? Refining this ratio into the form "error_of_current_method / error_of_initial_method * 100%" would clearly indicate how the particular simplification increased the error.

Author Response

Comments and Suggestions for Authors by the Reviewer

A detailed explanation of a narrowed practical example would be of great use (with all definitions, methods, procedures and results) - this will give obvious "touchable" understanding to the business/industry-oriented user. For example consider a detailed analysis of a partial problem of "maximizing the margin or minimizing fuel consumption for a power plant" (or another one) - maybe this worth writing a separate article on that to show not only academic but also "applied results" A more clear presentation of results comparison is required (for example - see table 1). What digits does the column "accuracy" actually show? 95% of what? Refining this ratio into the form "error_of_current_method / error_of_initial_method * 100%" would clearly indicate how the particular simplification increased the error.

Response to the Reviewer by the Authors

The authors would like to thank the reviewer for the very constructive comments. In our view the revised manuscript benefits greatly from the helpful hints. In the following our point-by-point response on the three identified suggestions is listed:

We appreciate this valuable remark and accordingly added a sub-chapter to our discussion sections: “Practical implications”. We hope that the described example helps more business-oriented users for getting a better idea of possible applications. We agree with the reviewer’s comment. Also in our view such a detailed example could be valuable but more suited for a follow-up, self-contained article. We tried to address this in the new sub-chapter „Practical implications“ by raising this as an additional point for the outlook. Thank you for this helpful comment. We have noticed for ourselves that the initial rough description of accuracy indicators in the continuous text in section 3.4.3 and above Table 7 was not sufficient. Therefore, we revised both the continuous text as well as the Table caption. In addition, as suggested by the reviewer, we provided an example for the calculation of an accuracy indicator in the section “Accuracy indicators”.

Reviewer 2 Report

The overall objective of the typescript was to systematically assess the effectiveness of the various performance-enhancing methods for ESL, which increase the calculation of technical conditions for the occurrence of the system performance: Product quality (power, electricity); Efficiency (economic, energy and ecological) of the processes of operation (processing, transmission, storage); Harmlessness of energy-processing products and processes.

The acceleration of the computation in the ESOM time scale and analysis in the direction of the permissible reduction of the model through aggregation and heuristic distribution methods has been highlighted. Analysis of energy systems was dependent on modelling tools, various and numerous models of energy system optimisation (ESOMs). To support decisions, models must be solvable within the allowable time span.

A novelty in these studies was the systematic evaluation of several reduction models and heuristic decomposition techniques for a large integrated (partly) energy system model using real data and focusing on maximum Speeds. The Model was used to study German energy scenarios, allowing for the expansion of storage capacity and transmission of electricity.

Computational time has been reduced by ten times, while maintaining accuracy. Explains the concept of "model reduction efficiency", provides opportunities for acceleration based on the memory of computers shared in the studies. These studies included methods for aggregating models at different scales and heuristic distribution strategies. Spatial calculations were realized (488 regions) and temporarily (8760 time steps) on the example of Germany's ESOM for the 2030 energy scenario.

The corresponding gains in performance depend on the breadness of the substantive model. "Model reduction efficiency" is possible and high, because the concept of decomposition is based on the solution of many thematic submodels: The decrease in computational time by model is less for very small submodels.

The article is a very long but accepted concept: the decomposition of known models, their descriptions and explanations, the broadband approach to optimization, the model of mathematical huge system and the lack of the formulation of the problem leading directly to the goal, solutions - Justify this length!

Author Response

We appreciate the reviewer’s comments. Especially the last point confirms our decision of not splitting the manuscript into a review paper and an experimental analysis in order to rather provide a comprehensive overview about the topic in a single article.

Reviewer 3 Report

In this paper, the authors have reported the findings of their systematic evaluation on various methods for modelling a large applied energy system, with the focuses on comparing the capabilities of several model reduction and heuristic decomposition techniques to accelerate the speed of search of the global optimum. The paper produced a detailed review, classification and comparison of model-based search accelerating, model reduction and heuristic/exact decomposition techniques.

Using German energy scenarios as case studies, several representative methods have been evaluated. The comparison and evaluation give a systematic assessment of model-based speed-up strategies performance, including computational efficiency and results accuracy.

The work is very valuable to those who work on the modelling and optimization of the large applied energy system. The materials have been well organized and presented, containing useful details.

The paper can be accepted with minor improvements.

Several suggestions are listed below:

In Section 4, detailed results and comparisons were presented for the computational efficiency and search accuracy improvements due to more effective model reduction schemes. It would be much more useful if a sensitivity analysis were added to illustrate the amount of improvement of different levels of model reduction.

Figure 6 to Figure 15 showed impressive results of model reduction with different resulting computation load and accuracy. An addition of sensitivity analysis to the overview table in Section 6 would be more beneficial to guide users to select a more appropriate technique.

Minor typos to be corrected: There are two Figure 7s, on pages 25 and 27. On page 27, section 4.2.1 – texts are centred. Several cross-references were missing the reference. The table in Section 6 should be Table 7, rather than Table 1.

Author Response

Comments and Suggestions for Authors by the Reviewer

In Section 4, detailed results and comparisons were presented for the computational efficiency and search accuracy improvements due to more effective model reduction schemes. It would be much more useful if a sensitivity analysis were added to illustrate the amount of improvement of different levels of model reduction. Figure 6 to Figure 15 showed impressive results of model reduction with different resulting computation load and accuracy. An addition of sensitivity analysis to the overview table in Section 6 would be more beneficial to guide users to select a more appropriate technique. Minor typos to be corrected: There are two Figure 7s, on pages 25 and 27. On page 27, section 4.2.1 – texts are centred. Several cross-references were missing the reference. The table in Section 6 should be Table 7, rather than Table 1.

Response to the Reviewer by the Authors

We appreciate these valuable remarks and very constructive comments. In our view the revised manuscript benefits greatly from these suggestions which we tried to address in the following:

We agree with this point of view. Although we already extensively discussed the value of an additional systematic sensitivity analyses in the initial version of our manuscript, we struggled whether this would overload the already extensive article. However, your comment motivated finding a compromise on this aspect: The revised version of the manuscript now contains an additional section in “Results” where the outcome of one exemplary sensitivity analysis is shown. Thank you for raising this point. As indicated above, we agree that the reliability of model-based scenario studies greatly benefits from conducting extensive sensitivity analyses. However, we deliberately decided to not do this in a systematic manner as, in our view, the major objective of the article is on identifying speed-up potentials which are related to computational indicators. As stated in section 5.2 of our manuscript, a variation of model inputs, which in a sense corresponds to sensitivity analyses, rather addresses the behavior of accuracy indicators. Considering the evaluation framework of our study, we are of the opinion that for computational indicators more general statements are feasible rather than for accuracy as the latter strongly depends on the analyzed use case (e.g. which cost relations exist; which of the several model outputs is really of interest; how far is a specific speed-up approach tuned to reduce accuracy errors of these outputs etc.). Nevertheless, as mentioned in section 5.4. of the article, we admit that there is potential for improvements in the future on this side. Still, we hope that the new exemplary sensitivity analysis in the revised manuscript supports guiding users when being confronted with the decision which of the analyzed speed-up approach to use. We appreciate these hints and corrected the affected parts of the manuscript. We also identified our way of PDF-building as one cause for the wrong figure captions.
